# LOCAL-CURVATURE-AWARE KNOWLEDGE GRAPH EMBEDDING VIA EXTENDED RICCI FLOW

## ABSTRACT

Knowledge graph embedding (KGE) relies on the geometry of the embedding space to encode semantic and structural relations. Existing methods place all entities on one homogeneous manifold—Euclidean, spherical, hyperbolic, or their product/multi-curvature variants, to model linear, symmetric, or hierarchical patterns. Yet a predefined, homogeneous manifold cannot accommodate the sharply varying curvature that real-world graphs exhibit across local regions. Since this geometry is imposed a priori, any mismatch with the knowledge graph's local curvatures will distort distances between entities and hurt the expressiveness of the resulting KGE. To rectify this, we propose **RicciKGE** to have the KGE loss gradient coupled with local curvatures in an extended Ricci flow such that entity embeddings co-evolve dynamically with the underlying manifold geometry towards mutual adaptation. Theoretically, when the coupling coefficient is bounded and properly selected, we rigorously prove that i) all the edge-wise curvatures decay exponentially, meaning that the manifold is driven toward the Euclidean flatness; and ii) the KGE distances strictly converge to a global optimum, which indicates that geometric flattening and embedding optimization are promoting each other. Experimental improvements on link prediction and node classification benchmarks demonstrate RicciKGE's effectiveness in adapting to heterogeneous knowledge graph structures.

## 1 INTRODUCTION

The manifold hypothesis (Fefferman et al., 2016) suggests that real-world data, despite residing in high-dimensional spaces, tend to concentrate near low-dimensional manifolds. This insight forms the theoretical foundation for a wide range of representation learning approaches, especially knowledge graph embedding. In KGE, entities and relations are mapped into continuous geometric embedding spaces, where distances and transformations are designed to capture the underlying semantic and structural patterns (Cao et al., 2024). Such embeddings have become the backbone of various reasoning tasks, including link prediction (Chami et al., 2020; Sun et al., 2019; Zhang et al., 2019), entity classification (Ji et al., 2015; Xie et al., 2016; Yu et al., 2022), and knowledge completion (Abboud et al., 2020; Zhang et al., 2021; 2020a).

Following the manifold hypothesis, the development of KGE methods increasingly focuses on identifying geometric spaces that better align with the underlying structure of knowledge graphs (Cao et al., 2024). Early KGE works primarily leveraged the Euclidean manifold due to its simplicity and translation invariance properties (Bordes et al., 2013). However, Euclidean embeddings cannot capture the non-linear structures prevalent in real-world knowledge graphs, such as hierarchical and cyclic patterns, due to their inherently curved geometry (Nickel & Kiela, 2017). To overcome this limitation, recent approaches have extended KGE to non-Euclidean manifolds, such as hyperbolic spaces for hierarchical relations (Balazevic et al., 2019; Chami et al., 2020;?; Wang et al., 2021) and spherical spaces for symmetric or periodic structures (Lv et al., 2018; Dong et al., 2021; Xiao et al., 2015). These advances better model specific structural motifs, but fall short of reflecting the rich local geometric heterogeneity in real knowledge graphs, which often comprise dense clusters, sparse chains, asymmetric relations, and layered hierarchies.

To model this complexity, more flexible embedding approaches have emerged, either employing dynamic curvature combinations (Yuan et al., 2023; Liu et al., 2024) or leveraging product manifolds

composed of multiple simpler geometries (Xiong et al., 2022; Zheng et al., 2022; Cao et al., 2022; Li et al., 2024), yet they still rely on a predefined and static geometric prior. This leads to a critical yet under-addressed issue:

*A homogeneous geometric prior, predefined before observing the data, forces embeddings to adapt to the "geometrically rigid" space, rather than shaping embedding space by real data manifold.*

Specifically, the core limitation lies in the use of a predefined manifold, which enforces uniform geometric constraints that fail to capture the localized curvature heterogeneity of real-world knowledge graphs. This misaligns the embedding space with the intrinsic manifold of the data, the optimal representation under the manifold hypothesis, resulting in degrading the embedding quality and impairing downstream reasoning (a comprehensive review of related work is provided in Appendix B.).

To resolve this misalignment, we propose **RicciKGE**, a curvature-aware KGE framework that couples the KGE loss gradient with the local discrete curvature (Fu et al., 2025; Naama et al., 2024) via an extended Ricci flow (Choudhury, 2024). Unlike static manifold-based methods, RicciKGE allows manifold geometry and entity embeddings to co-evolve in a closed feedback loop: curvature drives the metric to smooth curved regions and guide embedding updates, while the updated embeddings reshape curvature in return. This self-consistent process continuously aligns geometry with structure, eliminating the distortion induced by global geometric priors.

To theoretically ground this coupling, we establish convergence guarantees for the co-evolution process. Under mild geometric assumptions (Hebey, 2000; Saloff-Coste, 1994) and continuity/convexity conditions, we prove that all edge-wise Ricci curvatures decay exponentially to zero, thereby flattening the latent manifold toward a Euclidean geometry. Importantly, during this decay process, the local heterogeneity of curvature is not annihilated but gradually absorbed into the embedding updates through the extended Ricci flow, enabling entity embeddings to encode these geometric irregularities. Building on this absorption mechanism, we theoretically prove that the KGE objective still converges linearly to a global optimum under the evolving geometry. Since only entity embeddings remain learnable while curvature vanishes asymptotically, the final embeddings both preserve the imprints of local curvature variations and achieve global convergence.

Empirical results demonstrate that RicciKGE delivers consistent performance gains in two fundamental tasks. In link prediction, injecting RicciKGE's dynamic curvature flow into a variety of classical KGE models (Bordes et al., 2013; Yang et al., 2014; Sun et al., 2019; Chami et al., 2020; Li et al., 2024) with different distance functions yields universal improvements in WN18RR (Dettmers et al., 2018), FB15K-237 (Toutanova & Chen, 2015), and YAGO3-10 (Mahdisoltani et al., 2013), showing that the co-evolution of geometry and embedding is generalized between architectures. In node classification, RicciKGE surpasses both curvature-flow-only models (e.g. GNRF (Chen et al., 2025)) and standard message passing baselines (GCN (Kipf & Welling, 2016), GAT (Veličković et al., 2017)), demonstrating the advantage of jointly evolving curvature and embeddings. In addition, we visualize the convergence of curvature and embedding loss, providing empirical evidence for the theoretical convergence. All code will be made publicly available upon acceptance. In summary, we make the following contributions in this work.

- *Insight:* We expose the inherent limitation of predefined homogeneous geometries in KGE, which forces embeddings to fit a rigid space rather than adapt to real data manifold.
- *Method:* We propose **RicciKGE**, a curvature-aware framework coupling KGE optimization with extended Ricci flow, enabling the co-evolution of manifold geometry and entity embeddings in a closed loop.
- *Convergence:* We prove that RicciKGE achieves exponential curvature flattening and linear embedding convergence, effectively encoding local geometric heterogeneity into entity embeddings.

## 2 PRELIMINARIES

We briefly review the necessary preliminaries on knowledge graph embedding (KGE), Ricci curvature, and Ricci flow, providing the comprehensive background necessary for understanding our approach.

**Knowledge graph embedding.** A knowledge graph is defined as $\mathcal{G} = (\mathcal{V}, \mathcal{R}, \mathcal{T})$, where $\mathcal{V}$ is the entity set, $\mathcal{R}$ is the relation set, and $\mathcal{T} \subseteq \mathcal{V} \times \mathcal{R} \times \mathcal{V}$ is the set of factual triples. Knowledge graph

Figure 1: The framework of RicciKGE, which couples KGE gradients and local graph curvature in an extended Ricci flow, driving the latent manifold toward Euclidean flatness. The evolving geometry, in turn, guides curvature and gradient convergence and induces a closed-loop mechanism for bidirectional fitting between space and embedding.

embedding (KGE) methods learn low-dimensional representations $E(h), E(t) \in \mathbb{R}^d$ for entities and relation-specific transformations $f_r : \mathbb{R}^d \to \mathbb{R}^d$, such that a scoring function $\phi(E(h), f_r, E(t))$ assigns higher values to true triples than to corrupted ones. A general formulation is $\phi(h, r, t) := d\big(f_r(E(h)), E(t)\big)$, where $d(\cdot, \cdot)$ is a differentiable similarity or distance function. This unified view systematically covers the major classes of KGE models: translation-based methods such as TransE (Bordes et al., 2013), bilinear methods such as DistMult (Yang et al., 2014), geometric methods such as RotatE (Sun et al., 2019), and neural methods such as ConvE (Dettmers et al., 2018). The objective of KGE is typically based on a margin-ranking or logistic loss, e.g.,

$$\mathcal{L} = \sum_{(h,r,t) \in \mathcal{T}, (h',r,t') \notin \mathcal{T}} \big[\gamma + \phi(h, r, t) - \phi(h', r, t')\big]_+, \tag{1}$$

where $\gamma > 0$ is a margin hyperparameter and $[x]_+ = \max(0, x)$. Beyond Euclidean embeddings, KGE has been extended to Riemannian manifolds: hyperbolic spaces capture hierarchical relations, while spherical spaces model symmetric or cyclic patterns (Cao et al., 2024). However, their assumption of homogeneous curvature overlooks the heterogeneous local geometry of real-world knowledge graphs (Cao et al., 2024). To overcome this, we leverage Ricci curvature as a localized measure of distortion, which underpins our curvature-aware regularization.

**Ricci curvature on graphs.** Ricci curvature measures how neighborhood transport deviates from ambient distance, capturing local geometric irregularities. On graphs, the Ollivier–Ricci curvature (Ollivier, 2007; Hehl, 2024) of an edge $(i, j)$ with distance $d(i, j)$ is

$$\kappa(i, j) = 1 - \frac{W_1(\mu_i, \mu_j)}{d(i, j)},$$

where $W_1$ is the Wasserstein-1 distance between neighborhood measures $\mu_i$ and $\mu_j$. Positive curvature implies overlapping neighborhoods, while negative curvature indicates bottlenecks or tree-like expansions. Since knowledge graphs contain diverse relational motifs, their edge curvatures vary widely, making Ricci curvature a natural basis for curvature-aware KGE in non-uniform spaces.

**Ricci flow and its extension.** Given the heterogeneous local curvatures observed in knowledge graphs, a natural next step is to regulate their evolution through Ricci flow. Ricci flow (Hamilton, 1988) evolves a Riemannian metric by contracting positively curved regions and expanding negatively curved ones, progressively smoothing geometry over time. For graphs, we adopt its discrete variant (Naama et al., 2024), which updates edge weights according to local curvature, $w^{k+1}(i, j) = w^k(i, j)\big(1 - \frac{1}{2}\kappa^k(i, j)\big)$, followed by normalization. This process gradually reduces curvature disparities while preserving relative edge structures, and under mild conditions, converges to a stable metric, offering a tractable way to regularize knowledge graphs. However, discrete Ricci flow alone evolves independently of the KGE objective and does not directly influence entity representations. To overcome this limitation, we introduce an extended Ricci flow that:

$$\partial_t g_{ij} = -2 \operatorname{Ric}_{ij} + \beta \left(\nabla \mathcal{L} \otimes \nabla \mathcal{L}\right)_{ij}, \tag{2}$$

where $\beta$ is a coupling coefficient and $\mathcal{L}$ is the KGE loss. Here $g_{ij}$ denotes the components of the evolving metric tensor, which can be interpreted as the underlying edge-weight structure of the graph. $\operatorname{Ric}_{ij}$ is the Ricci curvature tensor capturing local geometric distortion. This extended formulation integrates curvature smoothing with gradient-based updates, ensuring that geometry evolution and representation learning proceed in tandem.

## 3 RICCIKGE

This section introduces the core mechanism, where entity embeddings learn seamlessly through a gradient-coupled Ricci flow: edge curvature evolution and embedding updates form a dynamic feedback loop, continuously adapting the geometry during training.

**Gradient-extended Ricci flow with learnable entity embedding.** Building upon the discrete (Naama et al., 2024) and extended (Choudhury, 2024; Lei & Baehr, 2025) Ricci flow frameworks, and inspired by embedding-based adaptations (Chen et al., 2025), we extend the classical distance update by replacing static nodes $(i, j)$ with dynamic KGE triplet embeddings $\left(E^{(k)}(h), f_r(\cdot), E^{(k)}(t)\right)$. Taking a forward-Euler step of size $\Delta s$ on the extended Ricci flow in Eq. 2, identifying $g_{ij} \equiv w_{ij}$, we get $w_{ij}^{k+1} = w_{ij}^k - \Delta s\, \kappa_{ij}^k\, w_{ij}^k + \Delta s\, \beta\, (\nabla \mathcal{L}^k \otimes \nabla \mathcal{L}^k)_{ij}$. Choosing $\Delta s = \frac{1}{2}$, which is the same in (Naama et al., 2024), yields the discrete extended Ricci-flow rule specialized to the KGE triple $(h, r, t)$ as follows:

$$w^{k+1}\left(E^{(k+1)}(h), f_r(\cdot), E^{(k+1)}(t)\right)$$

$$= w^k\left(E^{(k)}(h), f_r(\cdot), E^{(k)}(t)\right)\left(1 - \tfrac{1}{2}\,\kappa^k\left(E^{(k)}(h), E^{(k)}(t)\right)\right) + \tfrac{\beta}{2}\,\nabla_{E(h)}\mathcal{L}^k\,\nabla_{E(t)}\mathcal{L}^k, \quad (3)$$

where the curvature $\kappa^k\left(E^{(k)}(h), E^{(k)}(t)\right)$ is computed based on the current entity embeddings. Both the metric and the entity embeddings co-evolve under the discrete flow.

In this formulation, each relation $r$ is modeled as a learnable mapping $f_r(\cdot)$ through KGE optimization but remains static during the Ricci flow evolution. The reason for this choice is two-fold. One is *semantic invariance*. The relations encode abstract transformations that should remain consistent across graph regions. The preservation of invariant relation mappings is crucial to maintaining semantic consistency, as emphasized in (Lin et al., 2015). The other is *optimization stability*. Coupling the evolution of relations, entities, and curvature would make semantic learning entangle itself with geometric regularization, and thus destabilize training. By evolving only entity embeddings reflective of local structures, the model dynamically absorbs curvature heterogeneity across the entire knowledge graph while preserving global relational semantics, which is consistent with general intuitions in geometry-aware knowledge graph embedding (Lin et al., 2015).

**Entity embedding update.** The edge-weight function for our extended Ricci flow is defined as $w^k\left(E^{(k)}(h), f_r(\cdot), E^{(k)}(t)\right) := exp\left\{-d\left(f_r(E^{(k)}(h)), E^{(k)}(t)\right)\right\}$, for which a reason will be clear later. For simplicity, the loss function is defined as the distance $\mathcal{L}\left(E^{(k)}(h), f_r(\cdot), E^{(k)}(t)\right) := d\left(f_r(E^{(k)}(h)), E^{(k)}(t)\right)$. The entity embedding update can be formulated as a constrained optimization problem, where the discrete extended Ricci flow serves as a constraint. This problem can be solved using the method of Lagrange multipliers similar as in (Chen et al., 2025), yielding embedding updates that minimally perturb the geometry while complying with the flow dynamics. Let $a = \nabla_{E(h)} f_r\left(E(h)\right)^\top \nabla_{f_r(h)} d$ and $b = \nabla_{E(t)} d$. The edge-weight updating function can be written as: $w^{k+1} = w^k\left(1 - \tfrac{1}{2}\,\kappa^k\right) + \tfrac{\beta}{2}\,\langle a, b \rangle$. Since $w = e^{-d}$, the induced change in distance is:

$$\delta_d^k = -\frac{w^{k+1} - w^k}{w^k} = \tfrac{1}{2}\,\kappa^k - \frac{\beta}{2\,w^k}\,\langle a, b \rangle. \quad (4)$$

After that, the embedding update can be formulated as a constrained optimization problem using the method of Lagrange multipliers. Specifically, the embedding updates for the head and tail entities are defined as $\delta_h^k := E^{(k+1)}(h) - E^{(k)}(h)$ and $\delta_t^k := E^{(k+1)}(t) - E^{(k)}(t)$. The goal is to minimize the overall perturbation measured by the $L_2$ norm:

$$\min_{\delta_h^k, \delta_t^k} \|\delta_h^k\|^2 + \|\delta_t^k\|^2 \quad \text{s.t.} \quad \langle a, \delta_h^k \rangle + \langle b, \delta_t^k \rangle = \delta_d^k. \quad (5)$$

To ensure that the updated embeddings satisfy the discrete Ricci flow evolution of edge weights, we formulate the following Lagrangian:

$$\widehat{\mathcal{L}} = \|\delta_h^k\|^2 + \|\delta_t^k\|^2 + \lambda^k\left(\langle a, \delta_h^k \rangle + \langle b, \delta_t^k \rangle - \delta_d^k\right), \quad (6)$$

where $\lambda^k$ is the Lagrange multiplier. Solving the first-order optimality conditions, i.e., setting the partial derivatives of $\widehat{\mathcal{L}}$ with respect to $\delta_h$ and $\delta_t$ to zero, while minimizing the update norm, yields

---

**Algorithm 1:** RicciKGE Training with Discrete Distance Flow

---

**Input:** Triplets $\mathcal{T}$, entity embeddings $\mathbf{E}^k$ in $k$ iteration, graph $G$, loss $\ell(\cdot)$, curvature $\kappa(\cdot)$,
   relation mappings $f_r(\cdot)$ *(updated by the chosen KGE base method)*, coupling $\beta$

**Output:** Updated embeddings $\mathbf{E}^{(k+1)}$

**foreach** $(h, r, t) \in \mathcal{T}$ **do**

   Compute distance $d_{ht}^k = \|f_r(E^k(h)) - E^k(t)\|$;

   Weight $w_{ht}^k = \exp(-d_{ht}^k)$, curvature $\kappa_{ht}^k = \kappa(h, t, E, G, k)$;

   Step size $\eta_g^k = \beta/(2w_{ht}^k)$, gradient $g^k = \nabla_d \ell(d_{ht}^k)$;

   *Distance flow:* $d_{ht}^{k+1} = d_{ht}^k - \eta_g^k g^k + \frac{1}{2}\kappa_{ht}^k$; $\delta_d^k = d_{ht}^{k+1} - d_{ht}^k$;

   Compute gradients $\lambda^k \leftarrow -2\delta_d/(\|\nabla E^k(h)\|^2 + \|\nabla E^k(t)\|^2)$;

   Update: $E^{(k+1)}(h) = E^k(h) + \lambda^k \nabla E^k(h)$, $E^{(k+1)}(t) = E^k(t) + \lambda^k \nabla E^k(t)$;

**return** $\mathbf{E}^{(k+1)}$

---

$\delta_h^k = -\frac{\lambda^k}{2}\,a,\ \delta_t = -\frac{\lambda^k}{2}\,b$ which then leads to the closed-form expression for the multiplier $\lambda^k$:

$$\lambda^k = -\frac{2\,\delta_d^k}{\|a\|^2 + \|b\|^2} = -\frac{\kappa^k - \dfrac{\beta}{w^k}\,\nabla_{f_r(h)}d^\top\,\nabla_{E(h)}f_r\big(E(h)\big)\,\nabla_{E(t)}d}{\left\|G_h^{1/2}\,\nabla_{f_r(h)}d\right\|^2 + \left\|\nabla_{E(t)}d\right\|^2}, \qquad (7)$$

where $G_h := \nabla_{E(h)}f_r(E(h))\nabla_{E(h)}f_r(E(h))^\top$ is the local structure matrix capturing the transformation Jacobian of the head entity embedding. Until now, only one edge weight is taken into consideration. In fact, each entity $v$ typically participates in multiple triples in the knowledge graph, either as a head or tail entity. We define the set of all triples involving $v$ as $\mathcal{T}_v := \{(h, r, t) \in \mathcal{T} \mid v \in \{h, t\}\}$. To obtain a coherent embedding evolution, we aggregate the contributions from all incident triples, the final entity embedding updating as $\Delta E^k(v) = E^{(k+1)}(v) - E^{(k)}(v)$ is given by:

$$\Delta E^k(v) = \sum_{(h,r,t)\in\mathcal{T}} \lambda^k(h,t)\left(\mathbb{1}(v=h)\nabla_{E(h)}f_r(E(h))^\top\nabla_{f_r(h)}d + \mathbb{1}(v=t)\nabla_{E(t)}d\right), \qquad (8)$$

where $\mathbb{1}(\cdot)$ is the indicator function selecting the appropriate gradient contributions based on whether $v$ appears as head or tail. This unified formulation ensures that entity embeddings are updated along directions that simultaneously improve task-specific objectives and promote geometric regularity. The overall training procedure is summarized in Algorithm 1, which outlines the coupled distance flow and embedding update steps.

## 4 CONVERGENCE

**Curvature convergence** In the preceding subsection, we introduce RicciKGE's unified update mechanism, in which an extended Ricci-flow term adaptively refines edge weights to smooth local curvature, while a gradient-based term minimizes the triplet scoring objective to preserve global semantic coherence. Unlike static, predefined embedding spaces, this coupled evolution dynamically adapts to the geometry by jointly evolving entity embeddings and edge weights, allowing the model to faithfully reflect the underlying curvature of real-world knowledge graphs. To validate its effectiveness, we must demonstrate that under this iterative mapping, the Ricci curvature on every edge converges pointwise to zero, i.e., the manifold becomes asymptotically flat, thereby ensuring that all intrinsic curvature heterogeneity is faithfully captured in the learned entity embeddings. In what follows, we will establish a convergence theorem for the coupled iteration under assumptions of bounded geometry and weakly Lipschitz continuity, providing a rigorous theoretical foundation for the joint evolution of geometry (via Ricci flow) and semantics (via the distance). Detailed or missing proofs are deferred to the appendix D and E.

**Assumption 1** (Geometric and analytic regularity)**.** *The evolving metric space $(\mathcal{M}, g(s))$ satisfies the following for all $s \geq 0$: i) (Volume) $\mathrm{Vol}(\mathcal{M}) \geq V_0 > 0$; ii) (Diameter) $\mathrm{diam}(\mathcal{M}) \leq D < \infty$; iii) (Sobolev inequality[1]) for all $f \in \mathcal{C}^\infty(\mathcal{M})$, there exist a constant $C_\tau$ holding that: $\|f\|_{L^{\frac{2n}{n-2}}}^2 \leq$*

---

[1]$n = \dim\mathcal{M}$, $\mathcal{C}^\infty(\mathcal{M})$ stands for the space of real-valued functions on $\mathcal{M}$ that are differentiable to all orders (normale supérieure , France; Saloff-Coste, 1994).

$C_\tau \left( \|\nabla f\|_{L^2}^2 + V_0^{-1} \|f\|_{L^2}^2 \right)$, *and iv) (Spectral gap[2]) for any smooth tensor field $T$, $\int_{\mathcal{M}} \|\nabla T\|^2 \, dV \geq \lambda_1 \int_{\mathcal{M}} \|T\|^2 \, dV$, where $\lambda_1$ is the first non-zero eigenvalue of the (weighted) graph Laplacian $-\Delta$, also known as the* spectral gap.

This assumption provides a volume lower bound $V_0$ that prevents collapse, a uniform diameter bound $D$ that prevents blow-up, and a time-independent Sobolev constant that keeps the analytic control consistent. To proving curvature-flattening convergence, we present the following definitions.

**Definition 1** (Wasserstein–gradient Lipschitz constant). Let $\mathcal{F} : \mathcal{P}_2(\mathcal{M}) \to \mathbb{R}$ be a functional in the 2-Wasserstein space $\mathcal{P}_2(\mathcal{M})$ of Borel probability measures on $\mathcal{M}$ with finite second moments. Denote its Wasserstein gradient at $\rho \in \mathcal{P}_2(\mathcal{M})$ by $\nabla_W \mathcal{F}(\rho)$. We define

$$\mathcal{F}_W := \sup_{\rho_1 \neq \rho_2} \frac{\left\| \nabla_W \mathcal{F}(\rho_1) - \nabla_W \mathcal{F}(\rho_2) \right\|_{L^2(\mathcal{M})}}{W_2(\rho_1, \rho_2)}, \tag{9}$$

where $W_2(\rho_1, \rho_2)$ is the 2-Wasserstein distance between the two measures and $\left\| \cdot \right\|_{L^2(\mathcal{M})}$ is the $L^2$-norm taken with respect to the Riemannian volume element $dV$.

In our context, $\mathcal{F}_W$ controls the coupling strength between the geometry and the loss landscape, and thus drives the curvature-flattening process formalized below.

**Theorem 1** (Curvature-flattening convergence). *Let $\{(\mathcal{M}, g(s))\}_{s \geq 0}$ evolve under the discrete extended Ricci-KGE flow in Eq. 3.The curvature energy[3]ias defined as:$R(s) := \int_{\mathcal{M}} \|\mathrm{Ric}(g(s))\|^2 \, dV_{g(s)}$, which starts from an initial metric $g(0)$ whose curvature energy $R(0) := \int_{\mathcal{M}} \|Ric_{g(0)}(x)\|^2 \, dV_{g(0)} \leq K_0$ is finite and satisfying Assumption 1. If the coupling coefficient $\beta$ satisfies $0 < \beta < \frac{\lambda_1}{L_W^2 \sqrt{K_0}}$, then the discrete Ricci curvature converges pointwise to zero as $\max_{h,t} |\kappa_{ht}(s)| = 0$. Moreover, for any $\varepsilon > 0$, there exists a constant $C$ such that:*

$$\max_{h,t} |\kappa_{ht}(s)| \leq \varepsilon, \ \forall s \geq S(\varepsilon) := \frac{1}{\lambda_1 - \beta \mathcal{F}_W^2 \sqrt{K_0}} \log\left( \frac{2C\sqrt{K_0}}{\varepsilon} \right), \tag{10}$$

*which means that the curvature enters the $\varepsilon$-neighborhood of flatness in finite time $S(\varepsilon)$ with convergence rate $\mathcal{O}(\log(\frac{1}{\varepsilon}))$.*

*Proof sketch.* Differentiating under the coupled flow yields $\frac{dR}{ds} = -2\int_{\mathcal{M}} \|\nabla Ric\|^2 \, dV + 2\beta \int_{\mathcal{M}} \|Ric\| \|\nabla\mathcal{L}\|^2 \, dV$. Applying the Bochner (or Poincaré) inequality (Dai & Wei, 2012) $\int_{\mathcal{M}} \|\nabla Ric\|^2 \, dV \geq \lambda_1 \int_{\mathcal{M}} \|Ric\|^2 \, dV$, and the Lipschitz bound $\|\nabla\mathcal{L}\|_{L^2} \leq \mathcal{F}_W \sqrt{R}$ together with Young's inequality, one can show that $2\beta \int_{\mathcal{M}} \|Ric\| \|\nabla\mathcal{L}\|^2 \, dV \leq 2\beta \mathcal{F}_W^2 \sqrt{K_0} \, R(s)$. Hence, $\frac{dR(s)}{ds} \leq -C_r \, R(s)$, where $C_r = 2\lambda_1 - 2\beta \mathcal{F}_W^2 \sqrt{K_0}$. Grönwall's lemma (Gronwall, 1919) then gives $R(s) \leq R(0) \, e^{-C_r s}$. Invoking the uniform Sobolev inequality and a standard Moser iteration (Saloff-Coste, 2009) upgrades this to $L^2$-decay, $\|Ric\|_{L^\infty} \leq C \sqrt{K_0} \, e^{-C_r s/2}$. Finally, the manifold $\mathcal{M}$ is closed (i.e., compact and without boundary) and the uniform edge-weight bounds $|w_{ht}| \in [e^{-D}, 1]$ ensure the curvature vanishes in the limit. $\qquad\square$

Theorem 1 guarantees that the geometry induced by the coupled Ricci–KGE flow becomes asymptotically flat, progressively smoothing out edge-level curvature variations caused by non-uniform relational structures in real-world knowledge graphs. This flattening arises from a feedback loop where curvature guides embedding updates, and updated embeddings in turn reshape curvature, driving the co-evolution of geometry and representation toward consistency. While the coupled flow eventually drives the manifold toward a near-Euclidean regime, the curvature–gradient interactions that occur along the way are progressively absorbed into the embeddings. In this sense, RicciKGE captures heterogeneous structural signals during the transient phase, and the final flat geometry provides a stable and well-conditioned space for optimization. Since the optimization process is

---

[2]This inequality is a discrete analogue of the classical Bochner–Poincaré inequality, and it holds for any connected graph. In particular, Cheeger's inequality gives the lower bound $\lambda_1 \geq h^2/2$, where $h$ is the Cheeger constant of the graph (Spielman, 2012).

[3]Here and throughout, we let $\| \cdot \|$ denote the pointwise Hilbert–Schmidt norm of a tensor, e.g., $\|Ric\|^2 := \sum_{i,j} Ric_{ij}^2$, and $\| \cdot \|_{L^p}$ the corresponding $L^p$ norm over the manifold. In particular, $\| \cdot \|_{L^\infty}$ denotes the pointwise maximum (as $p \to \infty$).

intrinsically coupled with geometric evolution, it remains to examine whether the associated distance updates also converge, ensuring that curvature flattening indeed leads to stable metric optimization.

**Distance convergence** Building on the uniform curvature flattening from Theorem 1, we now relax the definition of loss by treating it as a convex function of distance, i.e., $\mathcal{L} = \ell(d)$ with $\ell(\cdot)$ convex. This relaxation is natural in the KGE setting, since most methods do not minimize the raw distance $d$ directly but instead optimize a convex transformation of it Eq. 1. Such a convexity assumption is standard in both KGE and convex optimization, and under this mild condition the update in Eq. 61 becomes a perturbed convex optimization, where the perturbation stems from an absolutely summable error $\sum_{k=0}^{\infty} |\kappa^k| < \infty$. For completeness, the precise requirements on $\beta$, convexity, and weight regularity are provided in Appendix E.1, leading to the following corollary.

**Corollary 1** (Linear convergence of distances). Under the assumptions of Theorem 1, for any edge $h \to t$, the distance sequence $\{d_{ht}^k\}_{k \geq 0}$ generated by the update rule equation 61 can be rewritten as

$$d_{ht}^{k+1} = d_{ht}^k - \eta_g^k \nabla_d \ell(d_{ht}^k) + \tfrac{1}{2} \kappa_{ht}^k, \ \eta_g^k := \frac{\beta}{2 w_{ht}^k}. \tag{11}$$

Assume that loss $\ell$ is $\mu$-strongly convex in $d$ and that the coupling coefficient $\beta$ satisfies $0 < \beta < \min\left\{ \frac{\lambda_1}{\mathcal{F}_W^2 \sqrt{K_0}}, \frac{4 e^{-D}}{\mu} \right\}$. Then there exists a contraction factor $q = \sup_k \left| 1 - \mu \eta_g^k \right| \in (0, 1)$ such that, for every $k \geq 0$ and any $\epsilon_d \geq 0$, it holds that $\left| d_{ht}^k - d_{ht}^\star \right| \leq q^k \left| d_{ht}^0 - d_{ht}^\star \right| + \tfrac{1}{2} \sum_{i=0}^{k-1} q^{k-1-i} \left| \kappa_{ht}^i \right| \leq \epsilon_d$. Consequently, due to $\sum_{i=0}^{\infty} |\kappa_{ht}^i| < \infty$, the sequence $\{d_{ht}^k\}$ converges to $d_{ht}^\star$ with linear rate.

The implications of the corollary can be understood from geometric and optimization perspectives. Geometrically, the distance-convergence result clarifies how the curvature-flattening phase triggered by discrete Ricci flow gradually hands control to the ordinary gradient optimization. Because Theorem 1 guarantees $\kappa_{ht}^k \to 0$ and $\sum_k |\kappa_{ht}^k| < \infty$, the curvature term in the distance update equation 11 acts only as a short-lived geometric correction; after finitely many iterations $T(\varepsilon)$ its influence disappears and the rule reduces to convex optimization on an almost flat manifold. Thus, Ricci flow first irons out heterogeneous curvature and then lets the gradient step refine distances, realizing a coherent transition from geometry-driven smoothing to Euclidean fine-tuning. From an optimization point of view, the residual $\kappa_{ht}^k$ behaves as a disappearing perturbation to distance convergence. Once $\kappa_{ht}^k$ decays, the step size stabilizes at $\eta_g^k \approx \beta/(2 w_{ht}^\star)$ and the iteration becomes $d_{ht}^{k+1} \approx d_{ht}^k - \eta_g^\star \nabla_d L(d_{ht}^k)$; the corollary then provides a linear contraction factor $q \in (0, 1)$. These findings show that discrete Ricci flow and gradient flow reinforce, rather than conflict with, each other: curvature evolution supplies global geometric regularization, while gradient descent secures the $O(q^k)$ convergence rate. Consequently, RicciKGE offers the first local-curvature-aware KGE framework with guaranteed linear convergence on a dynamically flattening manifold.

**Complexity considerations.** While the theoretical results above establish convergence, it is equally important to assess the computational overhead introduced by RicciKGE. Each epoch consists of two additional components beyond standard KGE updates. First, edge-wise curvature estimation involves computing a Sinkhorn distance between neighborhood distributions, with per-edge complexity $O(k^2 + kd)$ for average degree $k$ and embedding dimension $d$, yielding a total cost $O(|E|(k^2 + kd))$ per epoch. Second, Ricci-guided edge reweighting is realized through a lightweight Lagrangian update, costing $O(d)$ per edge, or $O(|E|d)$ overall. By contrast, classical KGE models such as TransE, DistMult, or RotatE typically require only $O(d)$ operations per triple. Thus, RicciKGE introduces an extra $O(k^2)$ term stemming from curvature estimation. However, this step is performed only once per epoch, is fully parallelizable, and in practice remains manageable since knowledge graphs are generally sparse ($k \ll d$). The added overhead is often offset by faster convergence in terms of both epochs and wall-clock time (see Appendix F.5), which makes RicciKGE computationally feasible for large-scale benchmarks.

## 5 EXPERIMENTS

**Datasets.** We evaluate our method on two types of graph representation tasks. For link prediction, which remains our core focus, we use three standard knowledge graph benchmarks with their canonical train/validation/test splits: WN18RR (Dettmers et al., 2018), FB15K-237 (Toutanova & Chen, 2015), and YAGO3-10 (Mahdisoltani et al., 2013). In addition, we include node classification to

Table 1: The comparison of typical KGE methods with application of our curvature evolution to the link prediction task using different types of distance function. W., Y., and F. present datasets of WN18RR, YAGO3-10, and FB15k-237.

| Model | | TransE | +Ours | DistMult | +Ours | RotatE | +Ours | AttH | +Ours | GoldE | +Ours |
|---|---|---|---|---|---|---|---|---|---|---|---|
| Distance | | $\|\mathbf{h} + \mathbf{r} - \mathbf{t}\|_2$ | | $\langle \mathbf{h}, \mathrm{diag}(\mathbf{r}), \mathbf{t}\rangle$ | | $\|\mathbf{h} \circ \mathbf{r} - \mathbf{t}\|_2$ | | $\|\mathbf{h} \oplus_{\mathbf{c}} \mathbf{r} \ominus_{\mathbf{c}} \mathbf{t}\|_{\mathbb{H}}$ | | $\|\mathbf{h} \circ \mathbf{r} - \mathbf{t}\|_{\mathcal{L}}^2$ | |
| Low embedding dimension 32 | | | | | | | | | | | |
| W. | MRR | 7.0 | **7.6** | 39.1 | **40.2** | 30.3 | **31.5** | 44.4 | **46.2** | 47.1 | **47.6** |
| | H@1 | 0.1 | **0.2** | 36.6 | **37.4** | 27.6 | **30.1** | 41.0 | **42.2** | 41.1 | **41.8** |
| | H@3 | 11.6 | **12.9** | 40.1 | **41.3** | 32.1 | **32.2** | 45.7 | **47.6** | 50.1 | **50.6** |
| | H@10 | 18.1 | **19.4** | 43.6 | **45.4** | 34.6 | **34.8** | 50.4 | **53.9** | 58.0 | 57.8 |
| F. | MRR | 29.3 | **31.0** | 28.5 | **28.9** | 28.5 | **28.8** | 31.4 | **31.9** | 32.4 | **32.7** |
| | H@1 | 20.7 | **22.1** | 20.2 | **20.8** | 20.4 | **20.7** | 22.4 | **23.0** | 23.7 | **24.1** |
| | H@3 | 32.2 | **34.3** | 31.1 | **31.6** | 31.1 | **31.4** | 34.4 | **35.1** | 35.4 | **35.7** |
| | H@10 | 46.7 | **49.0** | 45.3 | **45.5** | 44.7 | **45.1** | 49.5 | **49.8** | 49.7 | **50.1** |
| Medium embedding dimension 128 | | | | | | | | | | | |
| Y. | MRR | 30.4 | **33.9** | 17.3 | **18.0** | 39.4 | **40.6** | 36.7 | **37.4** | 39.5 | **40.8** |
| | H@1 | 20.4 | **23.3** | 10.8 | **11.6** | 29.7 | **30.0** | 25.7 | **26.5** | 30.0 | **30.8** |
| | H@3 | 35.4 | **39.7** | 18.7 | **19.1** | 44.4 | **46.2** | 41.7 | **42.5** | 43.3 | **46.6** |
| | H@10 | 49.4 | **53.4** | 29.9 | **30.5** | 57.9 | **60.3** | 58.7 | **59.2** | 58.2 | **59.8** |
| F. | MRR | 29.6 | **31.2** | 28.8 | **28.9** | 28.6 | **28.9** | 33.6 | **34.2** | 33.4 | **33.7** |
| | H@1 | 20.5 | **22.3** | 20.5 | **20.8** | 20.4 | **20.9** | 24.4 | **24.8** | 24.5 | **24.9** |
| | H@3 | 32.8 | **35.3** | 31.4 | 31.2 | 31.0 | **31.4** | 36.8 | **37.7** | 36.7 | **36.9** |
| | H@10 | 47.8 | **50.3** | 45.5 | **45.8** | 45.1 | **45.5** | 52.0 | **53.1** | 51.4 | **51.5** |
| High embedding dimension 256 | | | | | | | | | | | |
| F. | MRR | 30.2 | **31.7** | 29.0 | **29.7** | 29.2 | 29.1 | 33.8 | **34.3** | 34.1 | **34.2** |
| | H@1 | 21.1 | **21.7** | 20.9 | **21.6** | 21.2 | 21.1 | 24.5 | **25.0** | 25.1 | **25.2** |
| | H@3 | 33.3 | **35.9** | 32.0 | **32.2** | 31.7 | 31.5 | 37.0 | **37.8** | 37.3 | 37.3 |
| | H@10 | 48.5 | **50.9** | 45.8 | **46.0** | 45.4 | 45.4 | 52.6 | **53.2** | 52.0 | **52.1** |
| High embedding dimension 1000 | | | | | | | | | | | |
| W. | MRR | 22.4 | **22.8** | 43.8 | **44.2** | 47.5 | **48.1** | 46.6 | **46.8** | 51.3 | **51.6** |
| | H@1 | 1.4 | **1.7** | 39.3 | **39.9** | 42.7 | **43.7** | 40.8 | **41.1** | 46.3 | **46.9** |
| | H@3 | 40.2 | **40.6** | 45.0 | **45.6** | 49.4 | **49.7** | 49.3 | **49.5** | 53.3 | **53.4** |
| | H@10 | 53.0 | **53.4** | 53.4 | **53.6** | 57.3 | **58.0** | 57.4 | **57.5** | 60.7 | **60.9** |

test the robustness of curvature–gradient co-evolution beyond multi-relational KGs and to empirically support the general convergence discussed in section 4. For this purpose, we consider five large-scale graphs: Roman-Empire and Tolokers from the Heterophilous Graph Benchmark (Platonov et al., 2023), Cora_Full and Pubmed from CitationFull benchmark (Bojchevski & Günnemann, 2017), and OGBN-Arxiv from the Open Graph Benchmark (Hu et al., 2020). These datasets cover diverse graph structures and heterogeneous local curvature, providing complementary evidence for the generality of RicciKGE.

**Baselines** To comprehensively assess the effectiveness of our method on knowledge graph embedding (KGE), we compare against representative models from different families of distance function: TransE (Bordes et al., 2013) with translational distance, DistMult (Yang et al., 2014) with bilinear inner product, RotatE (Sun et al., 2019) with relation-specific rotation, AttH (Chami et al., 2020) with hyperbolic attention-based embeddings, and GoldE (Li et al., 2024) with universal orthogonal parameterization to generalize across product and mixed-curvature geometries. For node classification on attributed graphs, we benchmark against the classical GCN (Kipf & Welling, 2016), GAT (Veličković et al., 2017), and the recent curvature-driven model GNRF (Chen et al., 2025), where node embeddings evolve solely via the Ricci flow. More details on the experimental setting can be found in the Appendix F.

**Main results** The comparison results are reported in Table 1 and Table 2. For link prediction, RicciKGE consistently improves performance across all KGE models with different distance functions and embedding dimensions (32, 128, 256, 1000). For example, on WN18RR (dim=32), RicciKGE improves MRR by +0.55, +1.08, and +1.17 for TransE, DistMult, and RotatE, respectively. On FB15K-237 (dim=128), it yields relative gains of +0.16 for DistMult, +0.42 for RotatE, and +0.76 for AttH. Even on strong recent baselines such as AttH and GoldE, RicciKGE provides additional

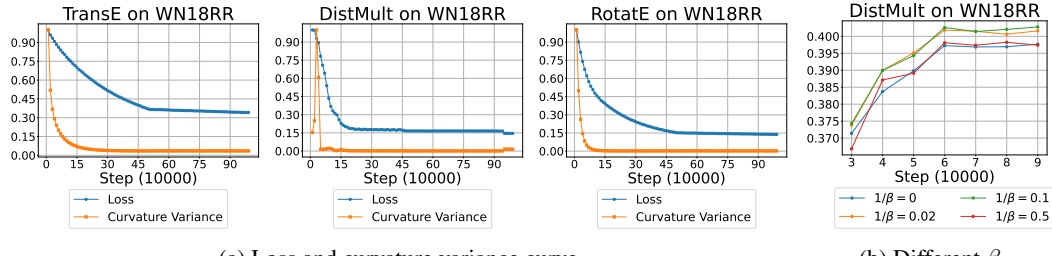

(a) Loss and curvature variance curve        (b) Different $\beta$

Figure 2: (a) Normalized loss and curvature variance curves on WN18RR (embedding dimension = 32). Curvature variance converges faster than loss, with both eventually stabilizing. (b) DistMult performance on WN18RR under varying coupling coefficients, showing that overly small or large values degrade effectiveness.

Table 2: Comparison of our methods with classical GNNs and SOTA Ricci-flow GRNF on the node classification task.

|        | Roman-Empire | Tolokers   | Core_Full  | Pubmed     | OBGN-Arxiv |
|--------|--------------|------------|------------|------------|------------|
| GCN,   | 71.23±0.22   | 79.61±0.66 | 68.06±0.98 | 86.74±0.47 | 66.34±0.76 |
| GAT    | 77.40±1.53   | 81.45±0.92 | 67.55±1.23 | 87.24±0.55 | 64.52±0.97 |
| GNRF   | 85.64±0.58   | 81.92±0.98 | 72.14±0.61 | 89.97±0.36 | 66.07±0.88 |
| Ours   | **87.40±0.54** | **82.54±0.87** | **72.51±0.53** | **90.57±0.42** | **67.25±0.76** |

gains, demonstrating its ability to adaptively flatten heterogeneous curvature beyond what static or pre-defined manifolds can capture. Although there are a very few cases where RicciKGE does not surpass the best baseline, the overall trend of consistent improvements across datasets and models remains clear, confirming that coupling Ricci flow with KGE training is beneficial in practice. For node classification, RicciKGE outperforms classical GNNs and the SOTA Ricci flow model GNRF on all datasets. For instance, it achieves $87.40\%$ accuracy on Roman-Empire versus $85.64\%$ from GNRF. This performance gap confirms that coupling curvature evolution with task gradients yields more effective representations than unsupervised geometric smoothing. By injecting task-aware feedback into the flow, RicciKGE aligns local geometry with global semantic objectives, resulting in more adaptive and discriminative embeddings.

**Convergence** Fig. 2a provides direct empirical evidence for our theoretical claim: both curvature variance and loss exhibit stable convergence, confirming the co-evolution of geometry and optimization. The faster convergence of curvature indicates that geometric regularization is completed earlier, which implies minimal perturbations introduced to gradient-based updates, and thus decoupled yet coordinated training dynamics.

**Coupling coefficient** Fig. 2b shows that the coupling coefficient significantly affects performance: small $\beta$ values underutilize curvature guidance, while overly large $\beta$ values distort gradient flow and degrade results. This agrees with our theory, which requires the coefficient to lie within a bounded range to ensure balanced co-evolution.

## 6 CONCLUSION

We address the geometric mismatch between predefined embedding manifolds and local heterogeneous structures of real-world knowledge graphs. We propose RicciKGE, a unified framework that couples KGE gradients with discrete Ricci curvature via an extended Ricci flow, enabling dynamic co-evolution of latent geometry and embeddings. Theoretically, we prove that RicciKGE drives all edge-wise curvature to zero, flattening the manifold toward Euclidean geometry, and guarantees convergence of edge distances from the embedding optimization. Empirically, RicciKGE achieves universal improvements in link prediction and node classification tasks.

**Limitations and future work** Our theoretically assumes a specific distance function of the form $d(f_r(E(h)), E(t))$, which limits the types of KGE models. This excludes models that use more complex or non-distance-based scoring functions. An interesting future work would be relaxing this assumption and extending our framework to more embedding models.

ETHICS STATEMENT

This work complies with the ICLR Code of Ethics and broader standards of responsible research. It does not involve human subjects, personally identifiable data, or other sensitive information that would require ethics approval. All datasets used are publicly available, properly licensed, and cited to their original sources. To promote transparency and reproducibility, we release our implementation along with detailed descriptions of our experimental settings. The research was carried out without conflicts of interest or external sponsorships that could have influenced its design, execution, or reporting.

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

APPENDIX CONTENTS

## A  NOTATIONS

For clarity, we summarize the main mathematical symbols used in our paper. We separate the notation into two parts: (i) symbols introduced in the **preliminaries and methods** (basic definitions, KGE modeling, curvature evolution, embedding update), and (ii) symbols specific to the **convergence analysis** (theoretical assumptions, curvature energy, contraction rates). This separation ensures that readers can easily locate the meaning of each symbol according to the section where it first appears.

Table 3 collects the symbols appearing in Sections 3.1 and 3.2 of the main text, including knowledge graph definitions, curvature-related terms, and the update rules in RicciKGE.

Table 4 collects the additional symbols appearing in Section 3.3 of the main text, where we establish the convergence guarantees. These symbols are mainly related to Riemannian geometry, Ricci flow dynamics, curvature energy, and contraction analysis.

Table 3: Notation used in the preliminaries and methods.

| Symbol | Meaning | Section |
|---|---|---|
| $\mathcal{G} = (\mathcal{V}, \mathcal{R}, \mathcal{T})$ | Knowledge graph (entities, relations, triples, edges) | Preliminaries |
| $\mathcal{V}$ | Set of entities (nodes) | Preliminaries |
| $\mathcal{R}$ | Set of relations | Preliminaries |
| $\mathcal{T}$ | Edge set of factual triples $(h, r, t)$ | Preliminaries |
| $\mathbf{E}(\mathcal{V}) \in \mathbb{R}^{|\mathcal{V}| \times d}$ | Entity embedding function | Preliminaries |
| $f_r(\cdot)$ | Relation-specific mapping function | Preliminaries |
| $h, r, t$ | Head entity, relation, tail entity of a triple | Preliminaries |
| $i, j$ | Coordinate indices | Preliminaries |
| $\delta_{ij}$ | Kronecker delta | Preliminaries |
| $N(i)$ | Neighbor set of node $i$ | Preliminaries |
| $z$ | Neighbor node of $i$ | Preliminaries |
| $w_{iz}$ | Edge weight between $i$ and $z$ | Preliminaries |
| $\delta_i(\cdot)$ | Dirac measure centered at node $i$ | Preliminaries |
| $d_{ht}$ | Distance between $h$ and $t$ under relation $r$ | Method |
| $w_{ht} = \exp(-d_{ht})$ | Edge weight for $(h, t)$ | Method |
| $\kappa_{ht}$ | Ricci curvature on edge $(h, t)$ | Method |
| $\beta$ | Coupling coefficient (gradient $\leftrightarrow$ curvature flow) | Method |
| $\eta_g$ | Gradient-based adaptive step size | Method |
| $\ell(d)$ | Loss function based on distance | Method |
| $g = \nabla_d \ell(d_{ht})$ | Gradient of loss w.r.t. distance | Method |
| $d_{ht}^{k+1}$ | Updated distance after Ricci flow | Method |
| $\delta_d$ | Distance update magnitude | Method |
| $a$ | Gradient direction w.r.t. head entity | Method |
| $b$ | Gradient direction w.r.t. tail entity | Method |
| $\lambda^k$ | Update multiplier for embeddings at iteration $k$ | Method |
| $E^{(k)}$ | Updated embedding after iteration $k$ | Method |
| $\mathbb{1}(\cdot)$ | Indicator function (head/tail selection) | Method |
| $k$ | Iteration index of Ricci flow update | Method |

Table 4: Notation used in the convergence analysis. The last column indicates where the symbol first appears (Assump./Def./Thm./Cor.).

| Symbol | Meaning | Where |
|---|---|---|
| $(\mathcal{M}, g(s))$ | Time-evolving Riemannian manifold and metric | Assump. |
| $\text{Vol}(\mathcal{M})$, $V_0$ | Volume of $\mathcal{M}$ and its positive lower bound | Assump. |
| $\text{diam}(\mathcal{M})$, $D$ | Diameter of $\mathcal{M}$ and its finite upper bound | Assump. |
| $C_\tau$ | Sobolev inequality constant on $\mathcal{M}$ | Assump. |
| $n = \dim \mathcal{M}$ | Dimension of the manifold | Assump. |
| $\nabla$, $\Delta$ | Gradient and (weighted) Laplacian on $\mathcal{M}$ | Assump. |
| $\lambda_1$ | Spectral gap (first non-zero eigenvalue of $-\Delta$) | Assump. |
| $T$ | Smooth tensor field on $\mathcal{M}$ | Assump. |
| $\int_{\mathcal{M}} \cdot \, dV$, $dV_{g(s)}$ | Integration over $\mathcal{M}$; volume element | Assump./Thm. |
| $\mathcal{P}_2(\mathcal{M})$ | 2-Wasserstein space of probability measures on $\mathcal{M}$ | Def. |
| $\rho$, $\rho_1$, $\rho_2$ | Probability measures in $\mathcal{P}_2(\mathcal{M})$ | Def. |
| $\nabla_W \mathcal{F}(\rho)$ | Wasserstein gradient of functional $\mathcal{F}$ at $\rho$ | Def. |
| $W_2(\rho_1, \rho_2)$ | 2-Wasserstein distance between $\rho_1$ and $\rho_2$ | Def. |
| $\| \cdot \|_{L^2(\mathcal{M})}$, $\| \cdot \|_{L^\infty}$ | $L^2$ / $L^\infty$ norms on $\mathcal{M}$ | Assump./Thm. |
| $\text{Ric}$, $\|\text{Ric}\|$, $\nabla\text{Ric}$ | Ricci tensor, its HS-norm, and its gradient | Thm./Sketch |
| $R(s)$ | Curvature energy $R(s) := \int_{\mathcal{M}} \|\text{Ric}(g(s))\|^2 \, dV_{g(s)}$ | Thm. |
| $K_0$ | Initial bound on curvature energy $R(0) \leq K_0$ | Thm. |
| $\beta$ | Coupling coefficient in extended Ricci flow | Thm./Cor. |
| $F_W$ | Wasserstein–gradient Lipschitz constant of the loss | Def./Thm. |
| $\kappa_{ht}(s)$ | (Discrete) Ricci curvature on edge $(h,t)$ at time $s$ | Thm. |
| $\max_{h,t} |\kappa_{ht}(s)|$ | Uniform edgewise curvature magnitude | Thm. |
| $C$ | Absolute constant appearing in decay-time bound | Thm./Sketch |
| $C_r$ | Decay rate constant $C_r = 2\lambda_1 - 2\beta F_W^2 \sqrt{K_0}$ | Sketch |
| $S(\varepsilon)$ | Entrance time into $\varepsilon$-flatness neighborhood | Thm. |
| $w_{ht}$ | Edge weight on $(h,t)$; bounded in $[e^{-D}, 1]$ | Sketch |
| $L = \ell(d)$, $\ell(\cdot)$ | Loss as a convex function of distance | Cor. |
| $\mu$ | Strong convexity parameter of $\ell(d)$ in $d$ | Cor. |
| $d_{ht}^k$, $d_{ht}^\star$ | Distance at iteration $k$ and its optimum | Cor. |
| $\eta_g^k$ | Step size $\eta_g^k := \beta/(2w_{ht}^k)$ | Cor. |
| $\kappa_{ht}^k$ | Discrete curvature on $(h,t)$ at iteration $k$ | Cor. |
| $q$ | Contraction factor $q = \sup_k |1 - \mu\eta_g^k| \in (0,1)$ | Cor. |
| $\varepsilon_d$ | Desired tolerance for distance error | Cor. |
| $T(\varepsilon)$ | Iteration/time after which curvature term is negligible | Cor./Text |

## B  RELATED WORK

**Manifold-based knowledge graph embedding**  Early manifold-based KGE methods utilized Euclidean space-based Cartesian coordinates, with models like TransE (Bordes et al., 2013) and RotatE (Sun et al., 2019) capturing simple relational patterns through translation and rotation, while PairRE (Chao et al., 2020) introduced more complex transformations. Polar coordinate-based methods, such as HAKE (Zhang et al., 2020b), and HBE (Pan & Wang, 2021), employed radial, angular, and Möbius transformations to capture hierarchical structures better. To embed hierarchical structures, hyperbolic models, including MuRP (Balazevic et al., 2019), ATTH (Chami et al., 2020), RotH (Chami et al., 2020), and RotL (Wang et al., 2021), leveraged the Poincaré ball and hyperbolic isometries for embeddings. In parallel, spherical models, such as TransC (Lv et al., 2018), HypersphereE (Dong et al., 2021), and ManifoldE (Xiao et al., 2015), utilized positive curvature to capture hierarchical and cyclic relations.

To address the limitations of single-geometry approaches in representing diverse knowledge graph (KG) structures, multi-space methods combine multiple geometric spaces or leverage dynamic mechanisms. Product space-based methods, such as UltraE (Xiong et al., 2022), HypHKGE (Zheng et al., 2022), GIE (Cao et al., 2022), and GoldE (Li et al., 2024) capture heterogeneous structures by integrating hyperbolic, spherical, and Euclidean spaces, while fusion-based approaches like

MCKG (Yuan et al., 2023) and UniGE (Liu et al., 2024) unify embeddings using adaptive optimization and transport techniques. Dynamic methods like DyERNIE (Han et al., 2020) and ODE-based models (Nayyeri et al., 2020) represent temporal and continuous relational patterns across geometries, and hybrid approaches like CurvRec (Wang et al., 2023) integrate mixed-curvature manifolds with Ricci-enhanced graph networks to model local and global properties effectively. Nevertheless, these multi-space methods still rely on rigid, prior-defined manifold geometries, restricting their ability to flexibly adapt to the heterogeneous and evolving structures. Our work attempts to break this limitation by introducing dynamic curvature evolution of the embedding space.

**Ricci flow in graph representation** Ricci flow has emerged as a powerful tool for geometric regularization in graph learning, evolving edge weights to flatten curvature and mitigate structural irregularities. Early approaches applied static Ricci curvature for graph rewiring to alleviate over-smoothing and Over-squashing (Liu et al., 2023; Sun et al., 2023). RicciNet (Sun et al., 2024) and Tian et al. (Tian et al., 2025) successfully applied the ricci flow for deep clustering. Fu et al. (Fu et al., 2025) integrate Ricci flow with information bottleneck principles to guide task-relevant information transport. Chen et al. (Chen et al., 2024) used Ricci flow-based differential equations to analyze the learning behavior of discretized neural networks. Recently, GNRF (Chen et al., 2025) and Ironing (Naama et al., 2024) implemented continuous and discrete Ricci flow to dynamically evolve graph geometry, with theoretical guarantees on curvature convergence and energy bounds. These advances motivate our investigation into the Ricci flow as a local-curvature-aware inductive bias for embedding knowledge graphs with heterogeneous relational structures.

## C  CURVATURE AND GRADIENT COUPLING FLOW-DRIVEN EMBEDDING EVOLUTION

### C.1  OBJECTIVE AND CONSTRAINT

We aim to update the entity embeddings by minimizing their perturbation while ensuring consistency with Ricci flow–induced changes in metric structure. This leads to a constrained optimization question. For a given triple $(h, r, t)$ in iteration $k$, we define:

$$
\begin{aligned}
\delta_h^k &= E^{(k+1)}(h) - E^{(k)}(h), \\
\delta_t^k &= E^{(k+1)}(t) - E^{(k)}(t),
\end{aligned}
\tag{12}
$$

where $E^{(k)}(h)$ and $E^{(k)}(t)$ are the embeddings of head and tail entities at iteration $k$. The goal is to minimize the overall perturbation magnitude:

$$
\min_{\delta_h^k, \delta_t^k} \|\delta_h^k\|^2 + \|\delta_t^k\|^2.
\tag{13}
$$

We require the update to respect the change in edge weight induced by the discrete extended Ricci flow as follows:

$$
\langle a, \delta_h^k \rangle + \langle b, \delta_t^k \rangle = \delta_d^k,
\tag{14}
$$

where

$$
\begin{aligned}
a &= \nabla_{E(h)} f_r(E(h))^\top \nabla_{f_r(h)} d, \\
b &= \nabla_{E(t)} d, \\
\delta_d^k &= \tfrac{1}{2}\kappa^k - \frac{\beta}{2w^k}\langle a, b \rangle.
\end{aligned}
$$

Here, $\kappa^k$ is the discrete Ricci curvature in iteration $k$, $\beta$ is the coupling coefficient between geometry and optimization, and $w^k = \exp(-d(f_r(E(h)), E(t)))$ is the edge weight defined via a soft exponential kernel over distance. The constraint reflects the Taylor expansion of the first order of the extended Ricci flow update rule applied to the edge $(h, r, t)$ and ensures that the change in evolved distance is consistent with the dynamics of the underlying curvature.

## C.2 RIGOROUS DERIVATION OF LINEAR CONSTRAINT

Before solving this constrained optimization question, our aim is to present the constraint equation 14 by combining a first-order expansion of the edge weight with the theoretical Ricci flow update. Specifically, for a given triple $(h, r, t)$, the edge weight is defined by:

$$w^k := \exp\left(-d\left(f_r(E^{(k)}(h)), E^{(k)}(t)\right)\right),$$

where $d(\cdot, \cdot)$ is a differentiable distance function, and $f_r(\cdot)$ is a relation-specific transformation. From the extended discrete Ricci flow update (Naama et al., 2024; List, 2008; Choudhury, 2024), the edge weight evolves as:

$$w^{k+1} = w^k\left(1 - \tfrac{1}{2}\kappa^k\right) + \tfrac{\beta}{2}\langle a, b\rangle.$$

We perturb the embeddings by Eq. equation 12:

$$E^{(k+1)}(h) = E^{(k)}(h) + \delta_h^k, \quad E^{(k+1)}(t) = E^{(k)}(t) + \delta_t^k.$$

Then the updated edge weight becomes:

$$w^{k+1} := \exp\left(-d\left(f_r(E^{(k)}(h) + \delta_h^k), E^{(k)}(t) + \delta_t^k\right)\right).$$

The original distance function is defined as follows:

$$d^k := d\left(f_r(E^{(k)}(h)), E^{(k)}(t)\right).$$

Let $x := f_r(E^{(k)}(h))$ and $y := E^{(k)}(t)$, so that: $d^k = d(x, y)$. We then perform a first-order expansion of the distance function under the perturbation:

$$d^{k+1} := d\left(f_r(E^{(k)}(h) + \delta_h^k),\ E^{(k)}(t) + \delta_t^k\right)$$
$$\approx d(x, y) + \left\langle \nabla_x d,\ \frac{\partial f_r}{\partial E(h)}\delta_h^k\right\rangle + \langle \nabla_y d,\ \delta_t^k\rangle$$
$$= d^k + \langle a, \delta_h^k\rangle + \langle b, \delta_t^k\rangle,$$

where we have

$$\nabla_x d := \nabla_{f_r(h)} d, \quad \nabla_y d := \nabla_{E(t)} d.$$

Thus, the distance change is:

$$\delta_d^k = d^{k+1} - d^k = \langle a, \delta_h^k\rangle + \langle b, \delta_t^k\rangle. \tag{15}$$

## C.3 LAGRANGIAN FORMULATION

To solve the constrained optimization problem, we introduce a Lagrange multiplier $\lambda^k \in \mathbb{R}$ and incorporate the constraint into the objective function, which is similar as (Chen et al., 2025). The resulting Lagrangian is defined as:

$$\mathcal{L}(\delta_h^k, \delta_t^k, \lambda^k) = \underbrace{\|\delta_h^k\|^2 + \|\delta_t^k\|^2}_{\text{embedding perturbation}} + \lambda^k \underbrace{\left(\langle a, \delta_h^k\rangle + \langle b, \delta_t^k\rangle - \delta_d^k\right)}_{\text{constraint residual}}. \tag{16}$$

This Lagrangian combines the smoothness objective (minimizing embedding changes) with a curvature-induced constraint that aligns embedding updates with Ricci flow geometry.

## C.4 SOLVING LAGRANGIAN VIA FIRST-ORDER OPTIMALITY

To find the stationary points of the Lagrangian, we compute the partial derivatives of $\mathcal{L}$ with respect to the variables $\delta_h^k$ and $\delta_t^k$. We first compute the gradient of $\mathcal{L}$ with respect to $\delta_h^k$:

$$\frac{\partial \mathcal{L}}{\partial \delta_h^k} = \frac{\partial}{\partial \delta_h^k}\left(\langle \delta_h^k, \delta_h^k\rangle + \lambda^k\langle a, \delta_h^k\rangle\right) = 2\delta_h^k + \lambda^k a.$$

Similarly, the gradient with respect to $\delta_t^k$ is:

$$\frac{\partial \mathcal{L}}{\partial \delta_t^k} = \frac{\partial}{\partial \delta_t^k} \left( \langle \delta_t^k, \delta_t^k \rangle + \lambda^k \langle b, \delta_t^k \rangle \right) = 2\delta_t^k + \lambda^k b.$$

To obtain the optimal solution for each triple $(h, r, t)$, we set the partial derivatives to zero and solve for the perturbations:

$$\frac{\partial \mathcal{L}}{\partial \delta_h^k} = 0 \quad \Rightarrow \quad 2\delta_h^k + \lambda^k a = 0 \quad \Rightarrow \quad \delta_h^k = -\frac{\lambda^k}{2} a, \tag{17}$$

$$\frac{\partial \mathcal{L}}{\partial \delta_t^k} = 0 \quad \Rightarrow \quad 2\delta_t^k + \lambda^k b = 0 \quad \Rightarrow \quad \delta_t^k = -\frac{\lambda^k}{2} b. \tag{18}$$

Substitute Eq.equation 17 and Eq.equation 18 into the constraint equation 15:

$$\langle a, \delta_h^k \rangle + \langle b, \delta_t^k \rangle = \delta_d^k$$

$$\left\langle a, -\frac{\lambda^k}{2} a \right\rangle + \left\langle b, -\frac{\lambda^k}{2} b \right\rangle = \delta_d^k$$

$$-\frac{\lambda^k}{2} \left( \|a\|^2 + \|b\|^2 \right) = \delta_d^k.$$

Solving for $\lambda^k$ gives:

$$\lambda^k = -\frac{2\delta_d^k}{\|a\|^2 + \|b\|^2}.$$

Next, we substitute the analytical form of $\delta_d^k$ into the expression for $\lambda^k$. Recall that under the extended discrete Ricci flow, the expected change in the logarithmic edge weight is given by:

$$\delta_d^k = \tfrac{1}{2} \kappa^k - \frac{\beta}{2w^k} \langle a, b \rangle.$$

Substituting this into the expression for $\lambda^k$ derived from the constraint, we obtain:

$$\lambda^k = -\frac{2\delta_d^k}{\|a\|^2 + \|b\|^2} = -\frac{2\left( \tfrac{1}{2}\kappa^k - \frac{\beta}{2w^k} \langle a, b \rangle \right)}{\|a\|^2 + \|b\|^2} = \frac{-\kappa^k + \frac{\beta}{w^k} \langle a, b \rangle}{\|a\|^2 + \|b\|^2}$$

$$= -\frac{\kappa^k - \frac{\beta}{w^k} \nabla_{f_r(h)} d^\top \nabla_{E(h)} f_r \left( E(h) \right) \nabla_{E(t)} d}{\left\| G_h^{1/2} \nabla_{f_r(h)} d \right\|^2 + \left\| \nabla_{E(t)} d \right\|^2}, \tag{19}$$

where $G_h := \nabla_{E(h)} f_r(E(h)) \nabla_{E(h)} f_r(E(h))^\top$ is the local structure matrix capturing the transformation Jacobian of the head entity embedding. Here the numerator computes the interaction between curvature gradient and embedding geometry; the denominator corresponds to a geometry-weighted norm controlling curvature flow strength. The entity embedding updates are linearly aligned with their local geometry gradient directions, scaled by a curvature-aware Ricci update term.

## C.5 Aggregated embedding update

We now derive the final form of the entity embedding update by aggregating all local updates over incident triples. Given a triple $(h, r, t)$, the optimal perturbations of the head equation 17 and tail equation 18 embeddings are:

$$\delta_h^k = -\frac{\lambda^k}{2} \cdot \nabla_{E(h)} f_r(E(h))^\top \nabla_{f_r(h)} d$$

$$\delta_t^k = -\frac{\lambda^k}{2} \cdot \nabla_{E(t)} d,$$

where $d = d(f_r(E(h)), E(t))$ is the distance function and $\lambda^k$ is the Lagrange multiplier. To compute the full update for an entity $v$, we aggregate contributions from all triples in which $v$ appears as either head or tail:

$$\Delta E^k(v) = \sum_{(h,r,t) \in \mathcal{T}_v} \lambda^k(h,t) \left( \mathbb{1}(v=h) \cdot \nabla_{E(h)} f_r(E(h))^\top \nabla_{f_r(h)} d + \mathbb{1}(v=t) \cdot \nabla_{E(t)} d \right). \quad (20)$$

Here, $\mathcal{T}_v := \{(h,r,t) \in \mathcal{T} \mid v = h \text{ or } v = t\}$ is the set of all triples incident to $v$, $\mathbb{1}(v = h)$ and $\mathbb{1}(v = t)$ are indicator functions determining the role of $v$, $\lambda^k(h,t)$ is as given in Eq.equation 19.

## D CURVATURE CONVERGENCE

### D.1 DEFINITIONS AND NOTATIONS: CURVATURE ENERGY AND CONTROL OBJECTIVE

To rigorously analyze the convergence behavior of the discrete Ricci-KGE flow, we begin by defining the curvature energy that monitors the flattening progress, clarify the norm conventions for tensor fields, and restate the control target under the geometric–analytic assumptions of the system.

**Curvature energy functional.** Let $(\mathcal{M}, g(s))$ be a family of Riemannian manifolds with evolving metrics governed by the discrete Ricci-KGE flow. We define the total Ricci curvature energy at time $s$ as:

$$R(s) := \int_{\mathcal{M}} \|\mathrm{Ric}(g(s))\|^2 \, dV_{g(s)}, \quad (21)$$

where 1) $\mathrm{Ric}(g(s))$ is the Ricci curvature tensor of the metric $g(s)$; 2) $dV_{g(s)}$ is the Riemannian volume form induced by $g(s)$; 3) $\| \cdot \|$ denotes the pointwise Hilbert–Schmidt norm defined below. This energy functional plays the role of a global geometric potential that the system seeks to minimize through a coupling with the KGE optimization (Hamilton, 1982; Chow & Knopf, 2004). The goal is to show that $R(s)$ decays to zero, ensuring that local curvature heterogeneity is eliminated over time:

$$\lim_{s \to \infty} R(s) = 0.$$

We emphasize that this energy is differentiable along the Ricci-KGE trajectory and admits a closed-form differential inequality along the flow (Hamilton, 1982), which forms the backbone of the convergence proof in the curvature convergence theorem. Throughout the paper, we employ the following norm conventions.

**(i) Hilbert–Schmidt norm (Lee, 2018) for tensors:** For any $(0, 2)$-tensor $T$, the pointwise norm is defined as:

$$\|T(x)\|^2 := \sum_{i,j} T_{ij}(x)^2,$$

which corresponds to the Frobenius norm under a local orthonormal frame.

**(ii) $L^p$-norms over the manifold:** For a tensor field $T$, the $L^p$-norm with respect to the evolving volume measure is:

$$\|T\|_{L^p(\mathcal{M})} := \left( \int_{\mathcal{M}} \|T(x)\|^p \, dV_{g(s)}(x) \right)^{1/p}.$$

Of particular interest are: i) $L^2$-norms, which arise in energy estimates and Bochner-type inequalities; ii) $L^\infty$-norms (Saloff-Coste, 2002), denoted $\|T\|_{L^\infty} := \sup_{x \in \mathcal{M}} \|T(x)\|$, used to bound maximum edgewise curvature during the flow.

### D.2 DERIVATIVE OF CURVATURE ENERGY

Firstly, we aim to compute the time derivative of the curvature energy functional equation 21, where both the integrand and the volume element depend on time $s$ through the evolving Riemannian metric $g(s)$. This requires carefully differentiating an integral over a time-varying manifold. For this, we use the *Leibniz rule on evolving manifolds* (e.g., Hamilton (1982)), which states that for any smooth function $f(s, x)$, it holds that:

$$\frac{d}{ds} \int_{\mathcal{M}} f(s,x) \, dV_{g(s)}(x) = \int_{\mathcal{M}} \left( \frac{\partial f}{\partial s} + f \cdot \frac{\partial}{\partial s} \log \sqrt{\det g(s)} \right) dV.$$

The first term corresponds to the pointwise derivative of the integrand, while the second term accounts for how the geometry of the underlying space is evolving. Noting that (Bailesteanu et al., 2010):

$$\frac{\partial}{\partial s} \log \sqrt{\det g} = \frac{1}{2} \mathrm{Tr}_g(\partial_s g),$$

we can rewrite the formula as:

$$\frac{d}{ds} \int_{\mathcal{M}} f(s,x) \, dV_{g(s)}(x) = \int_{\mathcal{M}} \left( \frac{\partial f}{\partial s} + \tfrac{1}{2} \mathrm{Tr}_g(\partial_s g) \cdot f \right) dV.$$

Additionally, letting $f(s,x) = \|\mathrm{Ric}(g(s))\|^2$, we obtain:

$$\frac{dR}{ds} = \int_{\mathcal{M}} \left( \frac{\partial}{\partial s} \|\mathrm{Ric}\|^2 + \tfrac{1}{2} \mathrm{Tr}_g(\partial_s g) \cdot \|\mathrm{Ric}\|^2 \right) dV. \tag{22}$$

This decomposition makes it possible to separately analyze the time derivative of the Ricci norm and the effect of volume change due to the evolving geometry.

**Computing the derivative of the Ricci norm.** We now compute the first term in Eq.equation 22, namely, the time derivative of the squared norm of the Ricci tensor. We begin with the expression:

$$\|\mathrm{Ric}\|^2 := g^{ik} g^{jl} \, \mathrm{Ric}_{ij} \, \mathrm{Ric}_{kl}, \tag{23}$$

where $g^{ij}$ denotes the inverse metric and $\mathrm{Ric}_{ij}$ is the Ricci curvature tensor. Differentiating Eq.equation 23 with respect to $s$, and applying the product rule, we obtain:

$$\frac{\partial}{\partial s} \|\mathrm{Ric}\|^2 = \frac{\partial}{\partial s} \left( g^{ik} g^{jl} \, \mathrm{Ric}_{ij} \, \mathrm{Ric}_{kl} \right) = 2 \left\langle \partial_s \mathrm{Ric}, \mathrm{Ric} \right\rangle + \left( \frac{\partial g^{ik}}{\partial s} g^{jl} + g^{ik} \frac{\partial g^{jl}}{\partial s} \right) \mathrm{Ric}_{ij} \mathrm{Ric}_{kl}. \tag{24}$$

The first term in Eq.equation 24 represents the leading-order contribution, as it directly measures how the curvature tensor evolves. The second term involves the time derivative of the inverse metric, which itself depends on $\partial_s g_{ij}$ through the identity:

$$\frac{\partial g^{ij}}{\partial s} = -g^{ik} g^{jl} \, \partial_s g_{kl}.$$

Thus, the remaining terms in Eq.equation 24 are nonlinear and of higher order in curvature and metric variation. In our analysis, we focus on the dominant part and write:

$$\frac{\partial}{\partial s} \|\mathrm{Ric}\|^2 = 2 \left\langle \partial_s \mathrm{Ric}, \mathrm{Ric} \right\rangle + \mathcal{R}_{\mathrm{high}},$$

where $\mathcal{R}_{\mathrm{high}}$ denotes the collection of higher-order remainder terms involving $\partial_s g^{ij}$. These terms are typically negligible in leading-order energy estimates and will be omitted in subsequent derivations. Hence we retain:

$$\frac{\partial}{\partial s} \|\mathrm{Ric}\|^2 \approx 2 \left\langle \partial_s \mathrm{Ric}, \mathrm{Ric} \right\rangle.$$

Plugging this Simplified formula equation 25 into Eq.equation 22 yields the main working form of the derivative of curvature energy:

$$\frac{dR}{ds} = 2 \int_{\mathcal{M}} \left\langle \partial_s \mathrm{Ric}, \mathrm{Ric} \right\rangle dV + \tfrac{1}{2} \int_{\mathcal{M}} \mathrm{Tr}_g(\partial_s g) \cdot \|\mathrm{Ric}\|^2 \, dV. \tag{25}$$

**Expanding the volume trace term** $\mathrm{Tr}_g(\partial_s g)$**.** We now compute the contribution from the change in the volume form, which enters via the trace (Topping, 2006) $\mathrm{Tr}_g(\partial_s g)$. Recall that $\mathrm{Tr}_g(\partial_s g) := g^{ij} \partial_s g_{ij}$, measures how the metric is expanding or contracting at each point. From extended Ricci flow, we have:

$$\partial_s g_{ij} = -2 \, \mathrm{Ric}_{ij} + \beta \, \nabla_i \mathcal{L} \nabla_j \mathcal{L}.$$

Compute the trace:

$$\begin{aligned}
\mathrm{Tr}_g(\partial_s g) &= g^{ij} \left( -2 \mathrm{Ric}_{ij} + \beta \nabla_i \mathcal{L} \nabla_j \mathcal{L} \right) \\
&= -2 g^{ij} \mathrm{Ric}_{ij} + \beta g^{ij} \nabla_i \mathcal{L} \nabla_j \mathcal{L} \\
&= -2 g^{ij} \mathrm{Ric}_{ij} + \beta \|\nabla \mathcal{L}\|^2,
\end{aligned}$$

where the second term is the squared norm of the loss gradient. Plugging this trace into Eq.equation 25 gives the volume correction term:

$$\int_{\mathcal{M}} \left( -2g^{ij}\mathrm{Ric}_{ij} + \tfrac{\beta}{2}\|\nabla\mathcal{L}\|^2 \right) \cdot \|\mathrm{Ric}\|^2 \, dV.$$

This term is a product of curvature and its square, hence cubic in nature. Although it is not always negligible, it is subdominant in our primary analysis and can be either omitted or bounded by higher-order remainder terms. Thus, the dominant contribution to $\frac{dR}{ds}$ comes from:

$$\frac{dR}{ds} \approx 2 \int_{\mathcal{M}} \langle \partial_s \mathrm{Ric}, \mathrm{Ric} \rangle \, dV. \tag{26}$$

### D.3 Deriving dominant evolution equation for curvature energy

In this step, we derive the leading-order evolution of the Ricci tensor $\mathrm{Ric}_{ij}$ under the extended Ricci flow (Choudhury, 2024; List, 2008; Lei & Baehr, 2025):

$$\partial_s g_{ij} = -2\,\mathrm{Ric}_{ij} + \beta\,\nabla_i\mathcal{L}\nabla_j\mathcal{L}.$$

This evolution contains two distinct contributions: 1) a geometric term $h_{ij}^{(\mathrm{geo})} = -2\,\mathrm{Ric}_{ij}$, and 2) a semantic coupling term $h_{ij}^{(\mathcal{L})} = \beta\,\nabla_i\mathcal{L}\nabla_j\mathcal{L}$. We compute their respective effects on the Ricci tensor by applying the standard variation formula. Firstly, let $h_{ij} := \partial_s g_{ij}$ be the variation of the metric. Then the variation of the Ricci tensor is given by (Chow & Knopf, 2004; Topping, 2006):

$$\partial_s\mathrm{Ric}_{ij} = \frac{1}{2}\left( \nabla^k\nabla_i h_{jk} + \nabla^k\nabla_j h_{ik} - \nabla^k\nabla_k h_{ij} - \nabla_i\nabla_j \mathrm{Tr}_g h \right) + Q_{ij}^{(\mathrm{curv})}, \tag{27}$$

where $Q_{ij}^{(\mathrm{curv})}$ denotes lower-order terms involving nonlinear contractions of the curvature tensor.

**Contribution from the geometric term:** We plug term $h_{ij}^{(\mathrm{geo})} = -2\mathrm{Ric}_{ij}$ into Eq.equation 27 to obtain:

$$\partial_s^{(\mathrm{geo})}\mathrm{Ric}_{ij}$$
$$= \frac{1}{2}\left( \nabla^k\nabla_i(-2\mathrm{Ric}_{jk}) + \nabla^k\nabla_j(-2\mathrm{Ric}_{ik}) - \nabla^k\nabla_k(-2\mathrm{Ric}_{ij}) - \nabla_i\nabla_j(-2R) \right) + Q_{ij}^{(\mathrm{curv})}$$
$$= -\left( \nabla^k\nabla_i\mathrm{Ric}_{jk} + \nabla^k\nabla_j\mathrm{Ric}_{ik} - \nabla^k\nabla_k\mathrm{Ric}_{ij} - \nabla_i\nabla_j R \right) + Q_{ij}^{(\mathrm{curv})}.$$

This is precisely the standard Ricci flow evolution expression. Its dominant contribution is the Laplace–Beltrami term (Hamilton, 1988):

$$\partial_s^{(\mathrm{geo})}\mathrm{Ric}_{ij} = \Delta\mathrm{Ric}_{ij} + (\text{nonlinear curvature terms}).$$

**Contribution from the semantic (gradient) term:** Next, substituting $h_{ij}^{(\mathcal{L})} = \beta\nabla_i\mathcal{L}\nabla_j\mathcal{L}$ into Eq.equation 27, we obtain:

$$\partial_s^{(\mathcal{L})}\mathrm{Ric}_{ij} = \frac{\beta}{2}\left( \nabla^k\nabla_i(\nabla_j\mathcal{L}\nabla_k\mathcal{L}) + \nabla^k\nabla_j(\nabla_i\mathcal{L}\nabla_k\mathcal{L}) - \nabla^k\nabla_k(\nabla_i\mathcal{L}\nabla_j\mathcal{L}) - \nabla_i\nabla_j\|\nabla\mathcal{L}\|^2 \right).$$

Note that the first three terms contain third-order derivatives of $\mathcal{L}$ and are thus higher-order terms. In the context of energy estimates, they are negligible compared to the fourth term, which is a symmetric second-order tensor. Therefore, we retain only the dominant term:

$$\partial_s^{(\mathcal{L})}\mathrm{Ric}_{ij} \approx -\frac{\beta}{2}\nabla_i\nabla_j\|\nabla\mathcal{L}\|^2.$$

Combining the dominant contributions from both the geometric and semantic terms, we obtain the main evolution equation:

$$\partial_s\mathrm{Ric}_{ij} = \Delta\mathrm{Ric}_{ij} + \frac{\beta}{2}\nabla_i\nabla_j\left(\|\nabla\mathcal{L}\|^2\right) + (\text{lower-order curvature terms}). \tag{28}$$

This expression captures the leading behavior of curvature evolution under the joint influence of intrinsic geometry and semantic coupling. Then, we substitute the dominant Ricci evolution Eq.equation 28 into the derivative of the curvature energy:

$$\frac{dR}{ds} := \frac{d}{ds} \int_{\mathcal{M}} \|\mathrm{Ric}\|^2 \, dV \approx 2 \int_{\mathcal{M}} \langle \partial_s \mathrm{Ric}, \mathrm{Ric} \rangle \, dV,$$

we obtain:

$$\frac{dR}{ds} \approx 2 \int_{\mathcal{M}} \left\langle \Delta \mathrm{Ric} + \frac{\beta}{2} \nabla^2 \left( \|\nabla \mathcal{L}\|^2 \right), \, \mathrm{Ric} \right\rangle dV. \tag{29}$$

We now separate the above into two interpretable terms: 1) The first term corresponds to the *geometric dissipation* driven by the Ricci flow:$2 \int_{\mathcal{M}} \langle \Delta \mathrm{Ric}, \mathrm{Ric} \rangle \, dV$, which promotes curvature smoothing. 2) The second term corresponds to the *semantic coupling* term driven by the gradient of the loss: $\beta \int_{\mathcal{M}} \langle \nabla^2 \left( \|\nabla \mathcal{L}\|^2 \right), \mathrm{Ric} \rangle \, dV$. Hence, the full expression becomes:

$$\frac{dR}{ds} \approx 2 \int_{\mathcal{M}} \langle \Delta \mathrm{Ric}, \mathrm{Ric} \rangle \, dV + \beta \int_{\mathcal{M}} \langle \nabla^2 \left( \|\nabla \mathcal{L}\|^2 \right), \mathrm{Ric} \rangle \, dV. \tag{30}$$

This decomposition cleanly isolates the geometric and semantic components, preparing for the energy dissipation estimate in the next step.

### D.4 GEOMETRIC DISSIPATION, SEMANTIC COUPLING, AND ENERGY DISSIPATION ESTIMATION

#### D.4.1 GRADIENT CONTROL VIA WASSERSTEIN–RICCI GEOMETRY

We aim to establish a bound on the Wasserstein gradient norm in terms of Ricci curvature energy.

**Definition 2** (Wasserstein–gradient Lipschitz constant). Let $\mathcal{F} : \mathcal{P}_2(\mathcal{M}) \to \mathbb{R}$ be a functional in the 2-Wasserstein space $\mathcal{P}_2(\mathcal{M})$ of Borel probability measures on $\mathcal{M}$ with finite second moments. Denote its Wasserstein gradient at $\rho \in \mathcal{P}_2(\mathcal{M})$ by $\nabla_W \mathcal{F}(\rho)$. We define:

$$\mathcal{F}_W := \sup_{\rho_1 \neq \rho_2} \frac{\|\nabla_W \mathcal{F}(\rho_1) - \nabla_W \mathcal{F}(\rho_2)\|_{L^2(\mathcal{M})}}{W_2(\rho_1, \rho_2)},$$

where $W_2(\rho_1, \rho_2)$ is the 2-Wasserstein distance between the two measures and $\|\cdot\|_{L^2(\mathcal{M})}$ is the $L^2$-norm taken with respect to the Riemannian volume element $dV$.

For any $\rho, \rho_0 \in \mathcal{P}_2$, choosing $\rho_0$ such that $\nabla_W \mathcal{F}(\rho_0) = 0$, we have that:

$$\|\nabla_W \mathcal{F}(\rho)\|_{L^2} \leq \mathcal{F}_W \cdot W_2(\rho, \rho_0). \tag{31}$$

Under the Ricci flow metric evolution $g(s)$, we consider the continuity equation $\partial_\theta \rho_\theta + \nabla \cdot (\rho_\theta \nabla \phi_\theta) = 0$. From the volume distortion induced by Ricci flow ( (Topping, 2009; Lott, 2006)), one obtains the key inequality:

$$\int_{\mathcal{M}} \rho_\theta \|\nabla \phi_\theta\|^2 \, dV_{g(s)} \leq \int_{\mathcal{M}} \phi_\theta \cdot \mathrm{Scal}(g(s)) \cdot \rho_\theta \, dV_{g(s)},$$

where $\mathrm{Scal}(g(s))$ is the scalar curvature at time $s$. By Hölder's inequality:

$$\int_{\mathcal{M}} \phi_\theta \, \mathrm{Scal} \, \rho_\theta \, dV \leq \left( \int_{\mathcal{M}} \phi_\theta^2 \, \rho_\theta \, dV \right)^{\frac{1}{2}} \left( \int_{\mathcal{M}} \mathrm{Scal}^2 \, \rho_\theta \, dV \right)^{\frac{1}{2}}. \tag{32}$$

and the Poincaré inequality:

$$\left( \int_{\mathcal{M}} \phi_\theta^2 \, \rho_\theta \, dV \right)^{\frac{1}{2}} \leq C_P \left( \int_{\mathcal{M}} \|\nabla \phi_\theta\|^2 \, \rho_\theta \, dV \right)^{\frac{1}{2}}.$$

we get following:

$$\int_{\mathcal{M}} \phi_\theta \, \mathrm{Scal} \, \rho_\theta \, dV \leq C_P \left( \int_{\mathcal{M}} \|\nabla \phi_\theta\|^2 \, \rho_\theta \, dV \right)^{\frac{1}{2}} \left( \int_{\mathcal{M}} \mathrm{Scal}^2 \, \rho_\theta \, dV \right)^{\frac{1}{2}}. \tag{33}$$

The dynamic formulation of Wasserstein distance yields:

$$W_2^2(\rho, \rho_0) \leq \int_0^1 \int_{\mathcal{M}} \rho_\theta \|\nabla \phi_\theta\|^2 \, dV_{g(s)} \, d\theta. \tag{34}$$

In orthonormal coordinates, scalar curvature is the trace of the Ricci tensor: Due to Cauchy–Schwarz inequality:

$$\mathrm{Scal}^2(g(s)) \leq \|\mathrm{Ric}(g(s))\|^2. \tag{35}$$

Let

$$u_\theta := \int_{\mathcal{M}} \|\nabla \phi_\theta\|^2 \, \rho_\theta \, dV, \qquad v_\theta := \int_{\mathcal{M}} \mathrm{Scal}^2 \, \rho_\theta \, dV.$$

By the dynamic formulation of the 2–Wasserstein distance,

$$W_2^2(\rho_\lambda, \rho_0) = \int_0^1 \int_{\mathcal{M}} \rho_\theta \, \|\nabla \phi_\theta\|^2 \, dV \, d\theta = \int_0^1 u_\theta \, d\theta. \tag{36}$$

For each $\theta \in [0, 1]$, Eq.equation 32 implies

$$u_\theta \leq \int_{\mathcal{M}} \phi_\theta \, \mathrm{Scal} \, \rho_\theta \, dV. \tag{37}$$

Applying Hölder's inequality in $L^2(dV)$ and then the Poincaré inequality (with respect to the same measure $\rho_\theta$) yields

$$\int_{\mathcal{M}} \phi_\theta \, \mathrm{Scal} \, \rho_\theta \, dV \leq \left( \int_{\mathcal{M}} \phi_\theta^2 \rho_\theta \, dV \right)^{1/2} \left( \int_{\mathcal{M}} \mathrm{Scal}^2 \, \rho_\theta \, dV \right)^{1/2} \leq C_P \, u_\theta^{1/2} \, v_\theta^{1/2}. \tag{38}$$

Combining Eq.equation 36–Eq.equation 38, we get:

$$\begin{aligned}
W_2^2 &\leq C_P \int_0^1 u_\theta^{1/2} v_\theta^{1/2} \, d\theta \\
&\leq C_P \left( \int_0^1 u_\theta \, d\theta \right)^{1/2} \left( \int_0^1 v_\theta \, d\theta \right)^{1/2} \qquad \text{(Cauchy–Schwarz in } \theta\text{)} \\
&= C_P \, W_2 \left( \int_0^1 v_\theta \, d\theta \right)^{1/2}.
\end{aligned} \tag{39}$$

Dividing by $W_2$ and squaring gives:

$$W_2^2 \leq C_P^2 \int_0^1 v_\theta \, d\theta. \tag{40}$$

Since Eq.equation 35,

$$\int_0^1 v_\theta \, d\theta = \int_0^1 \int_{\mathcal{M}} \mathrm{Scal}^2 \, \rho_\theta \, dV \, d\theta \leq \int_0^1 \int_{\mathcal{M}} \|\mathrm{Ric}\|^2 \rho_\theta \, dV \, d\theta. \tag{41}$$

Because the geodesic preserves total mass, $\int_0^1 \rho_\theta(x) \, d\theta = 1$ for a.e. $x \in \mathcal{M}$. Hence:

$$\int_0^1 \int_{\mathcal{M}} \|\mathrm{Ric}\|^2 \rho_\theta \, dV \, d\theta = \int_{\mathcal{M}} \|\mathrm{Ric}\|^2 \left( \int_0^1 \rho_\theta \, d\theta \right) dV = \int_{\mathcal{M}} \|\mathrm{Ric}\|^2 \, dV = R(s). \tag{42}$$

After that,putting Eq.equation 40, Eq.equation 41, and Eq.equation 42 together, we obtain:

$$W_2^2(\rho_\lambda, \rho_0) \leq C_P^2 \, R(s), \qquad \Longrightarrow \qquad W_2(\rho_\lambda, \rho_0) \leq C_P \, \sqrt{R(s)}. \tag{43}$$

Finally, plugging Eq.equation 43 into the Lipschitz control equation 31, we obtain:

$$\|\nabla_W \mathcal{F}(\rho)\|_{L^2} \leq \mathcal{F}_W C_P \cdot \sqrt{R(s)}.$$

Using the equivalence between Wasserstein and Euclidean gradient (Li & Li, 2018) norms:

$$\|\nabla\mathcal{L}\|_{L^2} = \|\nabla_W\mathcal{F}(\rho)\|_{L^2},$$

we obtain the desired estimate[4]:

$$\|\nabla\mathcal{L}\|_{L^2} \leq \mathcal{F}_W C_P \cdot \sqrt{R(s)}. \tag{44}$$

### D.4.2 GEOMETRIC DISSIPATION TERM

We first estimate the geometric dissipation term contribution to the curvature energy derivative $2\int_{\mathcal{M}}\langle\Delta\mathrm{Ric},\mathrm{Ric}\rangle\,dV$. For any smooth symmetric 2-tensor $T$, the Laplace–Beltrami operator $\Delta := \nabla^k\nabla_k$ satisfies:

$$\int_{\mathcal{M}}\langle\Delta T, T\rangle\,dV = -\int_{\mathcal{M}}\|\nabla T\|^2\,dV,$$

on closed manifolds or under appropriate boundary conditions. Applying this to $T = \mathrm{Ric}$, we obtain:

$$\int_{\mathcal{M}}\langle\Delta\mathrm{Ric},\mathrm{Ric}\rangle\,dV = -\int_{\mathcal{M}}\|\nabla\mathrm{Ric}\|^2\,dV.$$

Then, we apply a spectral gap estimate on the first eigenvalue (Chavel, 1984) of the Laplace operator:

$$\int_{\mathcal{M}}\|\nabla\mathrm{Ric}\|^2\,dV \geq \lambda_1\int_{\mathcal{M}}\|\mathrm{Ric}\|^2\,dV = \lambda_1 R(s),$$

where $\lambda_1 > 0$ denotes the first non-zero eigenvalue of the (weighted) graph or manifold Laplacian.

Substituting intogeometric dissipation term, we obtain:

$$2\int_{\mathcal{M}}\langle\Delta\mathrm{Ric},\mathrm{Ric}\rangle \leq -2\lambda_1 R(s). \tag{45}$$

### D.4.3 SEMANTIC COUPLING TERM

Next, we aim to control the semantic coupling term arising in the Ricci-KGE energy evolution. This term captures how the scalar potential $\mathcal{L}$ couples to the curvature of the manifold. Since $\mathcal{L} : \mathcal{M} \to \mathbb{R}$ is a smooth function, it holds that:

$$\|\nabla\mathcal{L}\|^2 = g^{ij}\partial_i\mathcal{L}\partial_j\mathcal{L}.$$

Taking the covariant Hessian (Hamilton, 1982; Topping, 2006) yields:

$$\nabla^2(\|\nabla\mathcal{L}\|^2) = \nabla_k\nabla_l(g^{ij}\partial_i\mathcal{L}\partial_j\mathcal{L}) \approx 2\nabla\mathcal{L}\otimes\nabla\mathcal{L}. \tag{46}$$

This approximation highlights the dominant contribution arising from the outer product of gradients. Substituting Eq.equation 46 into the curvature energy term equation 30, we get:

$$\beta\int_{\mathcal{M}}\langle\nabla^2(\|\nabla\mathcal{L}\|^2),\mathrm{Ric}\rangle\,dV \approx \beta\int_{\mathcal{M}}\langle2\nabla\mathcal{L}\otimes\nabla\mathcal{L},\mathrm{Ric}\rangle\,dV = 2\beta\int_{\mathcal{M}}\mathrm{Ric}(\nabla\mathcal{L},\nabla\mathcal{L})\,dV,$$

by the identity $\langle u\otimes u, T\rangle = T(u,u)$ for symmetric bilinear forms (Lee, 2018). We now estimate the right-hand side. By the norm inequality:

$$\mathrm{Ric}(\nabla\mathcal{L},\nabla\mathcal{L}) \leq \|\mathrm{Ric}\|\cdot\|\nabla\mathcal{L}\|^2,$$

we obtain:

$$2\beta\int_{\mathcal{M}}\mathrm{Ric}(\nabla\mathcal{L},\nabla\mathcal{L})\,dV \leq 2\beta\int_{\mathcal{M}}\|\nabla\mathcal{L}\|^2\cdot\|\mathrm{Ric}\|\,dV. \tag{47}$$

Applying Hölder's inequality (Saloff-Coste, 2002; Hebey, 2000) gives us:

$$\int_{\mathcal{M}}\|\nabla\mathcal{L}\|^2\cdot\|\mathrm{Ric}\|\,dV \leq \left(\int_{\mathcal{M}}\|\mathrm{Ric}\|^2 dV\right)^{1/2}\cdot\left(\int_{\mathcal{M}}\|\nabla\mathcal{L}\|^4 dV\right)^{1/2}, \tag{48}$$

---

[4]Throughout the derivation, various geometric constants (e.g., Poincaré inequality, Hölder bounds, trace-to-norm scaling) are omitted for clarity. These factors are ultimately absorbed into the effective constant $L_W$, which captures both the Wasserstein gradient Lipschitz coefficient and all structural inequalities in the bound.

where $R(s)$ denotes the Ricci energy, defined previously in Eq.equation 21. The first–order estimate has been proved as: $\|\nabla\mathcal{L}\|_{L^2} \leq \mathcal{F}_W \cdot C_P \cdot \sqrt{R(s)}$. For any smooth function $f$ on a compact 2–dimensional Riemannian manifold (Saloff-Coste, 2002; Hebey, 2000),

$$\|f\|_{L^4} \leq C_{GN}\|\nabla f\|_{L^2}^{1/2}\|f\|_{L^2}^{1/2},$$

Applying it with $f = \|\nabla\mathcal{L}\|$ gives

$$\|\nabla\mathcal{L}\|_{L^4}^4 \leq C_{GN}^4\left\|\nabla\|\nabla\mathcal{L}\|\right\|_{L^2}^2\|\nabla\mathcal{L}\|_{L^2}^2. \tag{49}$$

By the chain rule $\|\nabla\|\nabla\mathcal{L}\|\|_{L^2} \leq \|\nabla^2\mathcal{L}\|_{L^2}$. Ambrosio–Gigli–Savaré's *EVI $\Rightarrow$ Lipschitz* lemma (cf. (Ambrosio et al., 2008)) combined with the standard Bochner elliptic estimate yields the *Hessian–Lipschitz* bound[5]

$$\|\nabla^2\mathcal{L}\|_{L^2} \leq \mathcal{F}_W C_\Delta C_P \sqrt{R(s)}. \tag{50}$$

Insert Eq.equation 44 and Eq.equation 50 into Eq.equation 49, we get:

$$\|\nabla\mathcal{L}\|_{L^4}^4 \leq C_{GN}^4 (\mathcal{F}_W^2 C_\Delta^2 R(s))(\mathcal{F}_W^2 R(s)) = C_*^2 \mathcal{F}_W^4 R(s)^2, \qquad C_* := C_{GN}^2 C_\Delta C_P.$$

Unpacking the definition of the $L^4$ norm we arrive at

$$\int_\mathcal{M} \|\nabla\mathcal{L}\|^4 \, dV_{g(s)} \leq C_*\mathcal{F}_W^4 R(s)^2, \tag{51}$$

where all constants are explicit and depend only on $C_P, C_{GN}, C_\Delta$, and fixed geometric bounds of $\mathcal{M}$. Substituting the above into Eq.equation 48 yields:

$$\int \|\nabla\mathcal{L}\|^2 \cdot \|\mathrm{Ric}\| dV \leq C_*\mathcal{F}_W^2 R(s) \cdot \sqrt{R(s)} = C_*\mathcal{F}_W^2 R(s)^{3/2}. \tag{52}$$

Then plugging above into Eq.equation 47, absorbing $C_*$ into $\beta$, we obtain:

$$2\beta\int_\mathcal{M} \left\langle\nabla^2(\|\nabla\mathcal{L}\|^2), \mathrm{Ric}\right\rangle dV \leq 2\beta\mathcal{F}_W^2 R(s)^{3/2}. \tag{53}$$

Finally, if the coupling coefficient $\beta$ is small enough that $K_0 < \left(\frac{\lambda_1}{\beta\mathcal{F}_W^2}\right)^2$, we can get:

$$R(s)^{3/2} \leq \sqrt{K_0} \cdot R(s),$$

and thus:

$$\beta\int_\mathcal{M} \left\langle\nabla^2(\|\nabla\mathcal{L}\|^2), \mathrm{Ric}\right\rangle dV \leq 2\beta\mathcal{F}_W^2\sqrt{K_0} \cdot R(s). \tag{54}$$

### D.4.4 Exponential Decay of Curvature Energy

Finally, substituting the controlled geometric dissipation term equation 45 and the controlled gradient coupling term equation 54 into the derivative of the curvature energy equation 30, we obtain:

$$\frac{dR}{ds} \leq -2\lambda_1 R(s) + 2\beta\mathcal{F}_W^2\sqrt{K_0} \cdot R(s) = -C_r R(s),$$

where

$$C_r := 2\lambda_1 - 2\beta\mathcal{F}_W^2\sqrt{K_0}.$$

Here, the term $2\lambda_1$ arises from the Bochner-type spectral gap controlling the geometric dissipation of curvature, while the second term stems from the bounded semantic coupling contribution. The quantity $\mathcal{F}_W$ is the Wasserstein gradient Lipschitz constant of the objective function $\mathcal{L}$, and $K_0$ is the initial upper bound on the Ricci energy, i.e., $R(0) \leq K_0$. To ensure exponential decay, we require the overall coefficient $C_r$ of the curvature energy to remain positive, which imposes a restriction on the semantic coupling strength:

$$\beta < \frac{\lambda_1}{\mathcal{F}_W^2\sqrt{K_0}}.$$

---

[5]The constant $C_\Delta$ depends only on the geometric bounds of $(\mathcal{M}, g(s))$; see the main paper for its explicit expression.

This condition ensures that the dissipative effect from the Ricci curvature dominates any destabilizing effect introduced by the gradient interaction term. From the differential inequality:

$$\frac{dR}{ds} \le -C_r R(s),$$

we apply Grönwall's lemma Gronwall (1919) to get:

$$R(s) \le R(0)\,e^{-C_r s} \le K_0 \cdot e^{-(2\lambda_1 - 2\beta\mathcal{F}_W^2 \sqrt{K_0})s}. \tag{55}$$

This result establishes that the Ricci curvature energy $R(s) = \int_{\mathcal{M}} \|\mathrm{Ric}(g(s))\|^2 \, dV_{g(s)}$ decays exponentially over the course of the flow, under a bounded coupling strength $\beta$. Specifically, as $s \to \infty$, we obtain:

$$R(s) \to 0 \quad \text{exponentially fast.}$$

In geometric terms, this indicates that the underlying manifold evolves toward a flat state, with all Ricci curvature vanishing in the limit. Importantly, this flattening is not merely asymptotic but comes with an explicit convergence rate governed by $C_r$, which depends on the spectral geometry of the manifold (through $\lambda_1$), the initial curvature bound $K_0$, and the smoothness of the semantic objective $\mathcal{L}$ (through $\mathcal{F}_W$). This exponential decay plays a key role in guaranteeing stability of the Ricci-KGE dynamics and further supports convergence of the embeddings toward a consistent geometric structure.

### D.5 Upgrading $L^2$-decay to $L^\infty$-decay of Ricci curvature via Moser iteration

We have previously established that the Ricci curvature energy satisfies the exponential decay as Eq.equation 55, Our goal is to upgrade this $L^2$-decay into a pointwise $L^\infty$ estimate:

$$\|\mathrm{Ric}(g(s))\|_{L^\infty} \le C_\infty \sqrt{K_0}\, e^{-C_r s/2}.$$

Since Ricci curvature remains bounded in $L^\infty$ under the flow, we proceed by a standard three-step argument: applying the Sobolev inequality $\Rightarrow$ lifting to $L^p$ norms $\Rightarrow$ deriving an $L^\infty$ bound via Moser iteration (Saloff-Coste, 2009). We consider a compact Riemannian manifold $(\mathcal{M}, g(s))$ at any time $s$. Due to volume lower bound and diameter upper bound, the manifold admits a uniform Sobolev inequality:

$$\|f\|_{L^p}^2 \le C_S \left(\|\nabla f\|_{L^2}^2 + V_0^{-1}\|f\|_{L^2}^2\right), \quad p = \frac{2n}{n-2}.$$

Letting $f = |\mathrm{Ric}|$, and recalling that:

$$\|f\|_{L^2}^2 = R(s) \le K_0 e^{-C_r s},$$

we estimate $\|\nabla f\|_{L^2}^2$ [6] using the Bochner identity:

$$\frac{1}{2}\Delta|f|^2 = \|\nabla f\|^2 + \langle f, \Delta f\rangle + \text{lower-order terms.}$$

Using standard Ricci flow estimates and curvature evolution, we get:

$$\|\nabla f\|_{L^2}^2 \le C_P \|f\|_{L^2}^2, \tag{56}$$

where $C_P$ is the Poincaré-type constant (potentially dependent on $\lambda_1$) (Chavel, 1984; Li & Yau, 1980; Aubin, 1998). Substituting Eq.equation 56 into the Sobolev inequality:

$$\|f\|_{L^p}^2 \le C_S(C_P + V_0^{-1})\|f\|_{L^2}^2 := C_1\|f\|_{L^2}^2, \quad \Rightarrow \quad \|f\|_{L^p} \le C_1^{1/2}\|f\|_{L^2}.$$

Hence, $L^2$ decay implies $L^p$ decay:

$$\|f\|_{L^p} \le C_1^{1/2}\sqrt{K_0}e^{-C_r s/2} \to 0.$$

---

[6] $f = \|\mathrm{Ric}\|$ is the pointwise Hilbert–Schmidt norm, we recall that the curvature energy satisfies $\|f\|_{L^2}^2 = \int_{\mathcal{M}} \|\mathrm{Ric}\|^2 = R(s)$.

**Moser iteration from $L^p$ to $L^\infty$.** We define an increasing sequence of exponents:

$$q_0 = 2, \quad q_{k+1} = \frac{n}{n-2} q_k = \alpha q_k, \quad \alpha = \frac{n}{n-2} > 1.$$

Our goal is to control each $\|f\|_{L^{q_k}}$, and eventually obtain $\|f\|_{L^\infty}$. Assuming inductively that there exists $C_k$ such that:

$$\|f\|_{L^{q_k}} \leq C_k \|f\|_{L^2}.$$

To advance the iteration, let $u = f^{q_k/2}$, then:

$$\|f\|_{L^{q_{k+1}}}^{q_{k+1}} = \|u\|_{L^2}^2.$$

Applying Sobolev inequality:

$$\|u\|_{L^2}^2 \leq C_S \left( \|\nabla u\|_{L^2}^2 + V_0^{-1} \|u\|_{L^2}^2 \right).$$

Note that $\nabla u = \frac{q_k}{2} f^{q_k/2-1} \nabla f$, so:

$$\|\nabla u\|_{L^2}^2 = \left(\frac{q_k}{2}\right)^2 \|f^{q_k-2}\|_{L^\infty} \|\nabla f\|_{L^2}^2.$$

Using again Bochner control: $\|\nabla f\|_{L^2}^2 \leq C_P \|f\|_{L^2}^2$, and the inductive bound $\|f\|_{L^{q_k}} \leq C_k \|f\|_{L^2}$, we obtain:

$$\|f\|_{L^{q_{k+1}}} \leq C_{k+1} \|f\|_{L^2}, \quad C_{k+1} = \left[ C_S \left( \left(\frac{q_k}{2}\right)^2 C_P + V_0^{-1} \right) \right]^{1/q_k} C_k.$$

This recurrence has bounded growth since $\log C_k$ increases sublinearly while $q_k \to \infty$ exponentially. Therefore, there exists:

$$C_\infty := \sup_k C_k < \infty. \tag{57}$$

After that, taking the limit Eq.equation 57:

$$\|f\|_{L^\infty} = \lim_{k \to \infty} \|f\|_{L^{q_k}} \leq C_\infty \|f\|_{L^2},$$

and substituting $\|f\|_{L^2} \leq \sqrt{R(s)} \leq \sqrt{K_0} e^{-C_r s/2}$ yields:

$$\|\mathrm{Ric}(g(s))\|_{L^\infty} \leq C_\infty \sqrt{K_0} e^{-C_r s/2}. \tag{58}$$

Thus, we rigorously derive the $L^\infty$-decay estimate of $\mathrm{Ric}(g(s))$ via Moser iteration. The decay rate matches $\exp(-C_r s/2)$.

D.6   RICCI CURVATURE CONVERGENCE

Due to $(\mathcal{M}, g(s))$ is a smooth Riemannian manifold evolving under Ricci flow, where $s \geq 0$ is the time parameter. Since $\mathcal{M}$ is closed (i.e., compact and without boundary), any smooth tensor field defined over $\mathcal{M}$, such as $\mathrm{Ric}(g(s))$, is a continuous function with respect to $x \in \mathcal{M}$.

Therefore, for any $s \geq 0$, the pointwise norm

$$\|\mathrm{Ric}(g(s))(x)\| := \left( \sum_{i,j} \mathrm{Ric}_{ij}(x)^2 \right)^{1/2}$$

is a continuous function $x \mapsto \|\mathrm{Ric}(g(s))(x)\|$ on the compact manifold $\mathcal{M}$. Now recall that the $L^\infty$ norm of the Ricci tensor is defined as:

$$\|\mathrm{Ric}(g(s))\|_{L^\infty} := \sup_{x \in \mathcal{M}} \|\mathrm{Ric}(g(s))(x)\|.$$

By the hypothesis (from the previous curvature energy decay and Moser iteration), we have Eq.equation 58 as follows:

$$\|\mathrm{Ric}(g(s))\|_{L^\infty} \leq C_\infty \sqrt{K_0} e^{-C_r s/2}.$$

So, for every $x \in \mathcal{M}$, since the supremum is always greater than or equal to any pointwise value, we conclude:

$$\|\mathrm{Ric}(g(s))(x)\| \leq \|\mathrm{Ric}(g(s))\|_{L^\infty} \leq C_\infty \sqrt{K_0} e^{-C_r s/2}.$$

This pointwise upper bound on the Ricci tensor holds uniformly over the manifold and controls the decay of curvature in every direction. To relate the continuous curvature decay to the discrete edge-wise curvature $\kappa_{ht}(s)$[7] in RicciKGE, we assume a natural correspondence between the Riemannian metric tensor and the graph edge weights:

$$g_{ht}(s) \equiv w_{ht}(s),$$

i.e., the edge weight $w_{ht}(s)$ serves as a directional proxy for the metric component $g_{ht}(s)$. This assumption allows us to express metric evolution in the graph via the discrete extended Ricci flow:

$$w_{ht}^{(s+1)} = w_{ht}^{(s)} - \Delta s \cdot \kappa_{ht}^{(s)} \cdot w_{ht}^{(s)} + \Delta s \cdot \beta \, \nabla_i \mathcal{L}^{(s)} \nabla_j \mathcal{L}^{(s)}.$$

On the other hand, the continuous RicciKGE flow at the metric level evolves as:

$$\partial_s g_{ht}(s) = -2 \, \mathrm{Ric}_{ht}(s) + \beta \, \nabla_i \mathcal{L}(s) \nabla_j \mathcal{L}(s).$$

Substituting $g_{ht}(s) \equiv w_{ht}(s)$ and matching both expressions, we obtain:

$$\partial_s w_{ht}(s) = -2 \, \mathrm{Ric}_{ht}(s) + \beta \, \nabla_i \mathcal{L}(s) \nabla_j \mathcal{L}(s) = -\kappa_{ht}(s) \cdot w_{ht}(s) + \beta \, \nabla_i \mathcal{L}(s) \nabla_j \mathcal{L}(s),$$

from which we isolate the Ricci term and derive:

$$2 \, \mathrm{Ric}_{ht}(s) = \kappa_{ht}(s) \cdot w_{ht}(s).$$

In addition, the edge weights are assumed to obey the uniform bounds:

$$w_{ht}(s) \in [e^{-D}, 1] \quad \text{for all } i, j, s.$$

Combining with the pointwise Ricci decay

$$\|\mathrm{Ric}(g(s))\|_{L^\infty} \leq C_\infty \sqrt{K_0} e^{-C_r s/2},$$

we deduce that the discrete curvature satisfies:

$$|\kappa_{ht}(s)| \leq \frac{2 \, \|\mathrm{Ric}(g(s))\|_{L^\infty}}{w_{ht}(s)} \leq 2 e^D C_\infty \sqrt{K_0} \cdot e^{-C_r s/2}. \tag{59}$$

Hence:

$$\lim_{s \to \infty} \kappa_{ht}(s) = 0 \quad \text{for all } (i, j).$$

Moreover, for any $\varepsilon > 0$, inverting the exponential decay bound yields:

$$|\kappa_{ht}(s)| \leq \varepsilon \quad \text{for all } s \geq \frac{2}{C_r} \log\left( \frac{2 e^D C_\infty \sqrt{K_0}}{\varepsilon} \right). \tag{60}$$

Thus, the discrete curvature converges uniformly to zero under the extended RicciKGE flow.

## E   DISTANCE CONVERGENCE

Theorem of curvature convergence guarantees that the geometry induced by the coupled Ricci–KGE flow becomes asymptotically flat, progressively smoothing out edge-level curvature variations caused by non-uniform relational structures in real-world knowledge graphs. This flattening arises from a feedback loop where curvature guides embedding updates, and updated embeddings in turn reshape curvature, driving the co-evolution of geometry and representation toward consistency and adaptively regularizing the space into a unified Euclidean regime without external geometric priors. Since the optimization process is intrinsically coupled with geometric evolution, it remains to examine whether the associated distance updates also converge, ensuring that curvature flattening indeed leads to stable metric optimization. To formalize this link, we next make explicit the assumptions under which our convergence analysis holds.

---

[7]In the context of knowledge graph embedding (KGE), each fact is represented as a triple $(h, r, t)$, where $h$ and $t$ denote the head and tail entities, respectively. Accordingly, we use the notation $\kappa_{ht}$ instead of $\kappa_{ij}$ to emphasize that curvature is computed over entity pairs $(h, t)$ corresponding to the edges in knowledge graph triples.

### E.1 ASSUMPTIONS FOR CONVERGENCE ANALYSIS

**Spectral bound on $\beta$.** The coupling coefficient $\beta$ must be chosen below a spectral threshold to guarantee stability of the discrete Ricci flow. In particular, we require

$$0 < \beta \;<\; \min\left\{ \tfrac{4\lambda_1}{3F_W^2}, \; \tfrac{2\mu}{e^{K_0}}, \; \tfrac{\lambda_1}{\sqrt{K_0}} \right\},$$

where $\lambda_1$ is the first nonzero Laplacian eigenvalue of the graph, $F_W$ is a Lipschitz constant of the gradient field, $K_0$ is an upper bound on the initial curvature, and $\mu$ is the strong convexity parameter of the loss. This condition ensures that curvature dissipation dominates the perturbation from the gradient term.

**Convexity requirement.** The analysis assumes strong convexity of the loss with respect to the scoring gap $d(f_r(E(h)), g_r(E(t)))$. Note that we do not assume that $d(\cdot, \cdot)$ is a true metric: symmetry and the triangle inequality are unnecessary. It suffices that the scalar gap induces a convex landscape so that gradient descent admits a contraction argument.

**Regularity of weights.** We also assume bounded volume and diameter of edge weights during the discrete flow, which excludes degenerate cases where the manifold collapses.

**Remarks.** These assumptions are standard in geometric analysis and convex optimization. While margin-based losses used in practice are not strictly strongly convex, our experiments show that RicciKGE converges robustly even when the assumptions are relaxed. Moreover, empirical training curves remain stable when $\beta$ is set more aggressively than the theoretical bound, suggesting that the conditions are conservative rather than restrictive.

### E.2 DERIVATION OF DISTANCE UPDATE RULE AND LOSS RELAXATION JUSTIFICATION

Building on these assumptions, we next analyze how RicciKGE performs distance updates at the level of individual triples. At iteration $k$, each triple $(h, r, t)$ is characterized by a distance $d_{ht}^k$ whose evolution is governed by the edge weight dynamics induced by the discrete Ricci flow. To connect the geometric analysis with practical KGE training, we further express this distance through a loss function, ensuring consistency with standard optimization objectives. Concretely, RicciKGE links each distance $d_{ht}^k$ to an edge weight $w_{ht}^k$ via an exponential transformation:

$$w_{ht}^k := \exp(-d_{ht}^k) \quad \Longleftrightarrow \quad d_{ht}^k = -\log w_{ht}^k.$$

The induced change in distance is:

$$\delta_d^k = -\frac{w^{k+1} - w^k}{w^k} = \tfrac{1}{2}\kappa^k \;-\; \frac{\beta}{2\,w^k}\langle a, b\rangle. \tag{61}$$

Here, $a$ and $b$ are directional derivatives of the distance function:

$$a := \nabla_{E(h)} f_r(E(h))^\top \cdot \nabla_{f_r(h)} d, \quad b := \nabla_{E(t)} d,$$

where $f_r$ denotes the relation-specific transformation on entity embeddings. Rather than directly using the full gradient interaction $\langle a, b\rangle$, we approximate it by a scalar loss gradient. This is justified by:

$$\langle a, b\rangle = \left\langle \nabla_{E(h)} d, \nabla_{E(t)} d \right\rangle \approx \frac{\partial \ell(d)}{\partial d} = \nabla_d \ell(d_{ht}^k),$$

where $\ell(d)$ is a convex surrogate loss (e.g., margin ranking, logistic loss). This substitution ensures both theoretical tractability and compatibility with practical KGE objectives. Although RicciKGE evolves the underlying distance value $d_{ht}^k$, the actual training objective is not to directly minimize $d$, but rather to minimize a surrogate loss $\ell(d)$ built on top of it. This is motivated by several practical and theoretical reasons. First, most knowledge graph embedding (KGE) models—such as TransE, RotatE, and HousE—do not directly optimize the distance $d(h, r, t)$, but rather operate through a transformed loss function $\ell(d)$. For instance, margin-based ranking losses take the form:

$$\ell(d) = \max(0, d^+ - d^- + \gamma),$$

and probabilistic variants such as the logistic loss use:

$$\ell(d) = \log(1 + \exp(d)).$$

These losses are typically smooth, convex, and yield better convergence properties in optimization. Second, from a theoretical alignment perspective, since RicciKGE ultimately seeks to co-evolve geometry and embeddings under a unified objective, its update rule should reflect the actual training signal, i.e., $\ell(d)$, rather than raw distances. Third, by applying loss-level optimization, we can assume properties like $\mu$-strong convexity on $\ell(d)$, which is crucial for analyzing convergence behavior. The resulting distance update rule becomes:

$$d_{ht}^{k+1} = d_{ht}^k - \eta_g^k \cdot \nabla_d \ell(d_{ht}^k) + \tfrac{1}{2}\kappa_{ht}^k. \tag{62}$$

This reveals the evolution as a perturbed gradient descent step under a curvature-regularized dynamic geometry, where the Ricci term serves as a decaying geometric correction.

### E.3 SUMMABILITY OF CURVATURE PERTURBATION

Recall the curvature decay estimate stated in the main text. At Ricci flow time $s$, the edge curvature satisfies:

$$|\kappa_{ht}(s)| \leq \frac{2\|\mathrm{Ric}(g(s))\|_{L^\infty}}{w_{ht}(s)} \leq 2e^D C_\infty \sqrt{K_0} \cdot e^{-C_r s/2}.$$

Thus, the total curvature correction is summable:

$$\sum_{k=0}^\infty |\kappa_{ht}^k| \leq C_0 \sum_{k=0}^\infty e^{-\lambda k} = C_0 \cdot \frac{1}{1 - e^{-\lambda}} < \infty,$$

where $\lambda := \frac{C_r \delta}{2}$ and $C_0 := 2e^D C_\infty \sqrt{K_0}$. This completes the justification for the assumption used in following proof process as:

$$\sum_{k=0}^\infty |\kappa_{ht}^k| < \infty. \tag{63}$$

### E.4 DISTANCE CONVERGENCE TO OPTIMUM

We now study the convergence behavior of the RicciKGE distance sequence under the update rule equation 62 and the geometry-aware step size is defined as:

$$\eta_g^k := \frac{\beta}{2w_{ht}^k}, \quad \text{with } w_{ht}^k \in [e^{-D}, 1]. \tag{64}$$

From the curvature energy decay analysis in subsection E.3, we have justified the boundedness assumption $\sum_{k=0}^\infty |\kappa_{ht}^k| < \infty$. To ensure both convergence of the Ricci curvature energy and the stability of the distance update, we impose the following bound on the coupling parameter $\beta$. First, to guarantee convergence of the curvature dynamics, we require:

$$\beta < \frac{\lambda_1}{\mathcal{F}_W^2 \sqrt{K_0}}.$$

Second, in the distance update analysis, we rely on strong convexity of $\ell$ to yield a contraction in the update. To ensure this contraction is not overwhelmed by a large step size, we require:

$$\mu\eta_g^k < 1 \quad \Rightarrow \quad \mu \cdot \frac{\beta}{2w_{ht}^k} < 1 \quad \text{for all } k.$$

Since $w_{ht}^k \geq e^{-D}$, we conservatively require:

$$\beta < \frac{2}{\mu}e^{-D}.$$

Combining both requirements gives the final constraint on $\beta$:

$$0 < \beta < \min\left\{\frac{\lambda_1}{\mathcal{F}_W^2 \sqrt{K_0}}, \frac{2}{\mu}e^{-D}\right\}$$

Our goal is now to prove that under the assumptions of strong convexity, bounded step size as implied by the above $\beta$ constraint, and summable perturbation equation 63, the sequence $\{d_{ht}^k\}$ converges to the unique minimizer $d_{ht}^\star$. Instead of analyzing $d_{ht}^k$ directly, we define its deviation from the optimal value $d_{ht}^\star := \arg\min_d \ell(d)$ as:

$$r_{ht}^k := d_{ht}^k - d_{ht}^\star.$$

This transforms the convergence goal into showing that $|r_{ht}^k| \to 0$. We also rewrite the curvature term as an additive perturbation:

$$e_{ht}^k := \tfrac{1}{2}\kappa_{ht}^k, \quad \text{with } \sum_{k=0}^\infty |e_{ht}^k| < \infty.$$

Subtracting the optimal value $d_{ht}^\star$ from both sides of the update gives:

$$r_{ht}^{k+1} = d_{ht}^{k+1} - d_{ht}^\star = d_{ht}^k - \eta_g^k \nabla_d \ell(d_{ht}^k) + e_{ht}^k - d_{ht}^\star = r_{ht}^k - \eta_g^k (\nabla_d \ell(d_{ht}^k) - \nabla_d \ell(d_{ht}^\star)) + e_{ht}^k.$$

Note that $\nabla_d \ell(d_{ht}^\star) = 0$ by optimality. Since $\ell$ is assumed $\mu$-strongly convex and differentiable, its gradient satisfies:

$$\frac{\nabla_d \ell(d_{ht}^k) - \nabla_d \ell(d_{ht}^\star)}{d_{ht}^k - d_{ht}^\star} \geq \mu.$$

This implies:

$$\frac{\nabla_d \ell(d_{ht}^k) - \nabla_d \ell(d_{ht}^\star)}{r_{ht}^k} \geq \mu.$$

We define the contraction coefficient:

$$q_k := 1 - \eta_g^k \cdot \frac{\nabla_d \ell(d_{ht}^k) - \nabla_d \ell(d_{ht}^\star)}{r_{ht}^k} \quad \Rightarrow \quad |q_k| \leq 1 - \mu \eta_g^k.$$

Due to $0 < \beta < \frac{2}{\mu} e^{-D}$, then there exists $q < 1$ such that:

$$|q_k| \leq q < 1.$$

From the recurrence:

$$r_{ht}^{k+1} = q_k r_{ht}^k + e_{ht}^k,$$

we obtain the upper bound:

$$|r_{ht}^{k+1}| \leq q|r_{ht}^k| + |e_{ht}^k|.$$

Unrolling the recursion gives:

$$|r_{ht}^k| \leq q^k |r_{ht}^0| + \sum_{i=0}^{k-1} q^{k-1-i} |e_{ht}^i|.$$

Let $\epsilon := \sum_{i=0}^\infty |e_{ht}^i| < \infty$. For any $\varepsilon > 0$, choose $N$ large enough so that:

$$\sum_{i=N}^\infty |e_{ht}^i| < \varepsilon/2.$$

Split the perturbation sum as:

$$\sum_{i=0}^{k-1} q^{k-1-i} |e_{ht}^i| = \sum_{i=0}^{N-1} q^{k-1-i} |e_{ht}^i| + \sum_{i=N}^{k-1} q^{k-1-i} |e_{ht}^i|.$$

The first part satisfies:

$$\sum_{i=0}^{N-1} q^{k-1-i} |e_{ht}^i| \leq \max_{0 \leq i < N} |e_{ht}^i| \cdot \sum_{j=0}^{k-1-N} q^j = O(q^{k-N}) \to 0,$$

and the second is bounded by $\varepsilon/2$. Hence:

$$\limsup_{k \to \infty} |r_{ht}^k| \leq \varepsilon.$$

As $\varepsilon > 0$ can be chosen arbitrarily small, we conclude by definition of limit that:

$$\lim_{k \to \infty} |r_{ht}^k| = 0 \quad \Rightarrow \quad \lim_{k \to \infty} d_{ht}^k = d_{ht}^\star. \tag{65}$$

Therefore, under the combined assumptions of strong convexity, bounded geometry-aware step size, and summable perturbation, the sequence converges pointwise. We conclude:

$$\lim_{k \to \infty} d_{ht}^k = d_{ht}^\star \tag{66}$$

This completes the proof that the RicciKGE distance sequence converges precisely to the optimal point despite small curvature-induced noise.

## F    DETAILS OF EXPERIMENTS

### F.1    DATASET DETAILS

We evaluate our model on two types of graph representation tasks: link prediction and node classification. The datasets span diverse structures, scales, and curvature profiles, allowing a comprehensive assessment of our curvature-aware framework. Table 5and Table 6 summarize the key statistics and usage details for each dataset.

Table 5: Summary of link prediction datasets.

| Dataset | Entities | Relations | Train Triplets | Valid | Test |
|---------|----------|-----------|----------------|-------|------|
| WN18RR | 40943 | 11 | 86835 | 3034 | 3134 |
| FB15K-237 | 14541 | 237 | 272115 | 17535 | 20466 |
| YAGO3-10 | 123182 | 37 | 1079040 | 5000 | 5000 |

**Link prediction datasets**    These datasets are standard benchmarks for knowledge graph completion. We adopt the canonical train/validation/test splits as provided in the literature. For WN18RR (Dettmers et al., 2018) and FB15K-237 (Toutanova & Chen, 2015), the datasets are constructed to remove inverse relation leakage from their original counterparts (WN18 and FB15K). YAGO3-10 (Mahdisoltani et al., 2013) is a large-scale subset of the YAGO knowledge base, where entities with fewer than 10 associated relations are filtered out to ensure sufficient connectivity. Entities and relations are indexed and tokenized without additional preprocessing.

Table 6: Summary of node classification datasets.

| Dataset | Nodes | Edges | Classes |
|---------|-------|-------|---------|
| Roman-Empire | 22662 | 32927 | 18 |
| Tolokers | 11758 | 519000 | 2 |
| Cora_Full | 19793 | 64000 | 70 |
| Pubmed | 19717 | 44338 | 3 |
| OGBN-Arxiv | 169343 | 1166243 | 40 |

**Node classification datasets**    We evaluate on five large-scale graphs across homophilous and heterophilous regimes. For all datasets, we randomly shuffle nodes and split them into training/validation/test sets in a 60%/20%/20% ratio. Node features in Roman-Empire and Tolokersk (Platonov et al., 2023) are derived from word embeddings and platform metadata, respectively. Cora_Full and Pubmed (Bojchevski & Günnemann, 2017) use sparse bag-of-words TF/IDF features, while OGBN-Arxiv (Hu et al., 2020) uses dense word2vec-style embeddings of paper titles and abstracts. All graphs are treated as undirected except for OGBN-Arxiv, which retains its original directed citation edges. No additional normalization or feature scaling is applied beyond dataset-provided values.

## F.2 BASELINES AND EVALUATION METRICS

To fairly evaluate our method on knowledge graph embedding and node classification tasks, we compare against three KGE methods with different types of distance functions, and three different GNN methods.

### F.2.1 KNOWLEDGE GRAPH EMBEDDING BASELINES

**TransE** (Bordes et al., 2013) models a relation as a vector translation in embedding space. For a true triplet $(h, r, t)$, it enforces $\mathbf{h} + \mathbf{r} \approx \mathbf{t}$, where embeddings are constrained to lie in a low-dimensional Euclidean space. It captures one-to-one relations well but struggles with 1-to-N or N-to-N mappings.

**DistMult** (Yang et al., 2014) adopts a bilinear scoring function by modeling each relation as a diagonal matrix. The score for a triplet is computed as $f(h, r, t) = \langle \mathbf{h}, \mathbf{r}, \mathbf{t} \rangle$, where $\langle \cdot \rangle$ denotes tri-linear dot product. It is symmetric and thus cannot distinguish asymmetric relations.

**RotatE** (Sun et al., 2019)[8] embeds entities and relations into the complex vector space, interpreting each relation as a rotation from head to tail. Formally, $\mathbf{t} \approx \mathbf{h} \circ \mathbf{r}$, where $\mathbf{r} \in \mathbb{C}^d$ with modulus 1, and $\circ$ denotes element-wise complex multiplication. This enables modeling of symmetry, anti-symmetry, inversion, and composition.

**AttH** (Chami et al., 2020) embeds knowledge graphs in hyperbolic space to better capture hierarchical and power-law structures. It introduces an attention mechanism over relation-specific curvature, enabling different relations to adaptively select the appropriate geometry. The scoring function is defined by the hyperbolic distance between the transformed head and the tail, i.e., $f(h, r, t) = -d_{\mathbb{H}}(\mathbf{h} \oplus_{\mathbf{c}_r} \mathbf{r}, \mathbf{t})$, where $\oplus_{\mathbf{c}_r}$ denotes Möbius addition under relation-dependent curvature $\mathbf{c}_r$. By leveraging both hyperbolic geometry and attention, AttH effectively models hierarchical patterns and varying relation structures.

**GoldE** (Li et al., 2024) proposes a universal orthogonal parameterization framework that generalizes knowledge graph embedding across Euclidean, hyperbolic, and spherical spaces, as well as their product combinations. It learns orthogonal transformations that flexibly adapt to different curvature settings, making it capable of capturing heterogeneous relational patterns within a single unified model. The score function follows the Lorentzian distance form, $f(h, r, t) = -\|\mathbf{h} \circ \mathbf{r} - \mathbf{t}\|_{\mathcal{L}}^2$, where $\|\cdot\|_{\mathcal{L}}$ denotes the Lorentzian norm. GoldE achieves strong performance across diverse benchmarks by unifying multiple geometric assumptions under one parameterization.

### F.2.2 NODE CLASSIFICATION BASELINES

**GCN** (Kipf & Welling, 2016) performs spectral convolutions on graphs. Each layer aggregates feature information from immediate neighbors via a normalized Laplacian operator:

$$\mathbf{H}^{(l+1)} = \sigma \left( \tilde{D}^{-1/2} \tilde{A} \tilde{D}^{-1/2} \mathbf{H}^{(l)} \mathbf{W}^{(l)} \right),$$

where $\tilde{A} = A + I$, $\tilde{D}$ is its degree matrix, and $\sigma$ is a nonlinearity.

**GAT** (Veličković et al., 2017) generalizes GCNs by assigning attention weights to neighbors. The attention coefficient between nodes $i$ and $j$ is given by:

$$\alpha_{ij} = \mathrm{softmax}_j \left( \mathrm{LeakyReLU}(\mathbf{a}^\top [\mathbf{W}\mathbf{h}_i \| \mathbf{W}\mathbf{h}_j]) \right).$$

**GNRF** (Chen et al., 2025)[9] is a curvature-driven model where node embeddings evolve via Ricci flow. The edge weights $w_{ij}(t)$ adapt over time based on curvature $\kappa_{ij}(t)$, and node features evolve as:

$$\frac{d\mathbf{h}_i}{dt} = \sum_{j \sim i} -\kappa'_{ij}(t) \left[ \mathbf{h}_j - \cos(\mathbf{h}_j, \mathbf{h}_i)\mathbf{h}_i \right].$$

---

[8]We extend the implementation from `https://github.com/DeepGraphLearning/KnowledgeGraphEmbedding` for our link prediction tasks.

[9]We extend the implementation from `https://github.com/Loong-Chan/GNRF_new` for our node classification tasks.

### F.2.3 Evaluation metrics

**Link prediction.** We use filtered Mean Reciprocal Rank (MRR) and Hits@K (K=1, 3, 10) as evaluation metrics:

$$\text{MRR} = \frac{1}{N} \sum_{i=1}^{N} \frac{1}{\text{rank}_i}, \quad \text{Hits@K} = \frac{1}{N} \sum_{i=1}^{N} \mathbb{I}(\text{rank}_i \leq K),$$

where $\text{rank}_i$ is the position of the correct entity and $\mathbb{I}$ is the indicator function.

**Node classification.** We evaluate performance using standard classification accuracy, defined as:

$$\text{Accuracy} = \frac{|\{v \in \mathcal{V}_{\text{test}} : \hat{y}_v = y_v\}|}{|\mathcal{V}_{\text{test}}|},$$

where $\hat{y}_v$ denotes the predicted label for node $v$, $y_v$ is the ground-truth label, and $\mathcal{V}_{\text{test}}$ is the set of test nodes.

### F.3 Implementation details

All experiments are implemented in Python 3.9 and executed on a Linux server equipped with an Intel(R) Xeon(R) Gold 6240R CPU @ 2.40GHz, 256 GB of RAM, and four NVIDIA RTX 4090 GPUs (24 GB VRAM each), running CUDA 12.2 and driver version 535.230.02. Our code is based on PyTorch 2.3.1. For node classification, we use PyTorch Geometric 2.6.1.

### F.4 Hyperparameter settings

We organize hyperparameter settings into three parts: knowledge graph embedding (KGE), Ricci flow coupling, and node classification.

**Knowledge graph embedding.** For KGE experiments on WN18RR, FB15K-237, and YAGO3-10, all baselines are trained with a margin-based ranking loss using batch size 512, 1024 negative samples per positive triple, and up to 100k training steps. Optimization is performed with Adam and early stopping on validation MRR. **TransE** adopts margin values of 6.0 (WN18RR) and 24.0 (YAGO3-10), with learning rates of $5 \times 10^{-5}$ and $2 \times 10^{-4}$, respectively. **DistMult** uses slightly larger learning rates and adds L2 regularization. **RotatE** is trained in the complex space with the same margins and learning rates as TransE. **AttH** and **GoldE** follow their original hyperparameter settings, with learning rates tuned per dataset. For all models, test batch sizes are adjusted per dataset to fit GPU memory.

**Ricci flow configuration.** Ricci flow is invoked every 5 epochs to update edge weights. Unless otherwise specified, we fix the curvature-task coupling coefficient $\beta$ to 0.1.

**Node classification.** We evaluate on five datasets: Roman-Empire, Tolokers, Cora_Full, Pubmed, and OGBN-Arxiv, each trained for 500 epochs (1000 for OGBN-Arxiv) using the GNRF framework and an implicit ODE solver. The curvature-feature coupling coefficient $\beta$ is set to 0.01. The ODE integration interval $[10^{-5}, t_1]$ is dataset-specific, with $t_1$ ranging from 1.56 to 7.18. Hidden dimensions are set to 256 or 64 depending on dataset size, and dropout rates range from 0.09 to 0.56. Learning rates vary from $6.8 \times 10^{-4}$ to $1.1 \times 10^{-2}$, with Adam optimizer and tuned weight decay. Batch normalization is applied at both input and output layers, except for OGBN-Arxiv where only output normalization is used. Adjoint integration is enabled only for OGBN-Arxiv to reduce memory usage.

### F.5 Efficiency experiment

We report both per-epoch runtime and total wall-clock time to reach the same target MRR in Table 7. While RicciKGE increases the per-epoch time (e.g., 14.05s → 22.65s on WN18RR, and 25.8 → 41.8 minutes per 10k steps on FB15k-237), the number of epochs required for convergence is significantly reduced (29 → 14 and 55k → 30k, respectively). As a result, the total time-to-target MRR decreases by 22% on WN18RR and 12% on FB15k-237, demonstrating that Ricci-guided updates improve

Table 7: Runtime comparison of RicciKGE vs. vanilla backbones. Although per-epoch time is higher, the total time-to-target MRR is reduced due to faster convergence.

| Dataset | Model | Epoch Time | Epochs to Target MRR | Total Time |
|---------|-------|------------|----------------------|------------|
| WN18RR | TransE | 14.05s | 29 | 407s |
| WN18RR | TransE + Ricci | 22.65s | 14 | 317s |
| FB15k-237 | GoldE | 25.8 min / 10k steps | 55,000 | 141.9 min |
| FB15k-237 | GoldE + Ricci | 41.8 min / 10k steps | 30,000 | 124.5 min |

---

**Algorithm 2:** RicciKGE with Discrete Distance Flow and Structured Embedding Evolution

---

**Input:** Triplets $\mathcal{T}$, entity embeddings $\mathbf{E}$, graph structure $G$, relation mappings $f_r(\cdot)$, loss $\ell(d)$, Ricci curvature $\kappa(\cdot)$, coupling coefficient $\beta$

**Output:** Updated entity embeddings $\mathbf{E}^{(k+1)}$

**foreach** *entity* $v \in \mathcal{E}$ **do**
    Initialize update accumulator: $\Delta E^k(v) \leftarrow 0$

**foreach** *triplet* $(h, r, t) \in \mathcal{T}$ **do**
    Compute $f_r(E(h))$, $d_{ht} = \|f_r(E(h)) - E(t)\|$;
    Compute edge weight: $w_{ht} = \exp(-d_{ht})$;
    Compute curvature: $\kappa_{ht} = \texttt{Ricci}(h, t, E, G)$;
    Compute gradient-based step size: $\eta_g = \frac{\beta}{2w_{ht}}$;
    Compute loss gradient: $g = \nabla_d \ell(d_{ht})$;
    *Distance evolution:*
$$d_{ht}^{k+1} = d_{ht} - \eta_g \cdot g + \tfrac{1}{2}\kappa_{ht}$$
$$\delta_d = d_{ht}^{k+1} - d_{ht}$$
    Compute:
$$a := \nabla_{E(h)} f_r(E(h))^\top \cdot \nabla_{f_r(h)} d, \quad b := \nabla_{E(t)} d$$
$$\lambda^k = \frac{-2\,\delta_d}{\|a\|^2 + \|b\|^2}$$
    Accumulate embedding updates: $\Delta E^k(h) \mathrel{+}= \lambda^k \cdot a$, $\Delta E^k(t) \mathrel{+}= \lambda^k \cdot b$

**foreach** *entity* $v \in \mathcal{E}$ **do**
    Update embedding: $E^{(k+1)}(v) \leftarrow E^{(k)}(v) + \Delta E^k(v)$

**return** $\mathbf{E}^{(k+1)}$

---

optimization efficiency in practice. These results validate our theoretical convergence analysis and highlight RicciKGE's scalability advantages under realistic wall-clock budgets.

### F.6 ALGORITHM DESCRIPTION

Algorithm 2 outlines the structured update mechanism used in RicciKGE. For each triplet $(h, r, t)$, we first compute the distance between the projected head entity $f_r(E(h))$ and the tail entity $E(t)$, followed by the edge weight via an exponential kernel. The Ricci curvature $\kappa_{ht}$ is computed with respect to the graph geometry and current entity embeddings. The distance is then evolved through a discrete forward step that combines a task-driven gradient descent term with a curvature correction term, as given by the Ricci-extended flow update rule.

To preserve alignment between geometry and optimization, we compute the induced change in distance $\delta_d$, and solve for the scalar step coefficient $\lambda^k$ using a minimal-perturbation formulation. This ensures that entity updates follow the least-effort direction while maintaining consistency with the flow dynamics. Gradients are computed with respect to both the head and tail embeddings, structured via the Jacobian of the relation mapping $f_r(\cdot)$. The total embedding update for each entity is accumulated across all incident triples. At the end of each iteration, the embeddings are synchronously updated, enabling co-evolution of geometry and representation in a unified framework.

### F.7 Extended Results

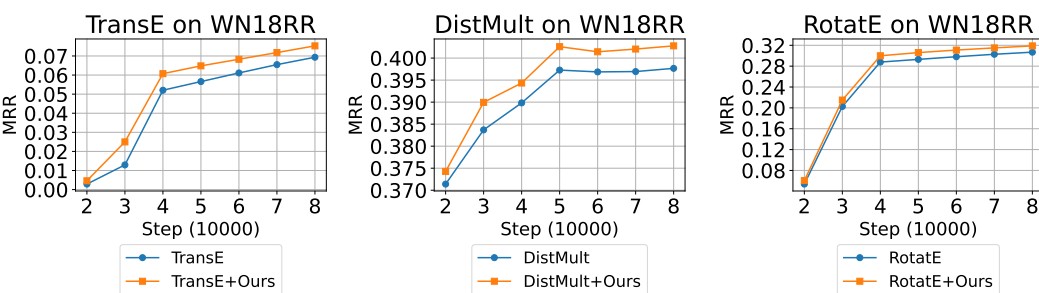

Figure 3: Performance comparison of baseline KGE models and RicciKGE-enhanced variants on WN18RR over training steps. Blue curves represent the vanilla baseline models, while yellow curves denote models trained with our RicciKGE extension.

Figure 3 shows the MRR performance of three representative knowledge graph embedding models (TransE, DistMult, RotatE) on the WN18RR dataset during training. We compare the vanilla models against their RicciKGE-enhanced counterparts, where discrete Ricci flow is integrated to guide geometry-aware optimization. Each curve tracks the MRR across training steps, with horizontal axes representing step counts (in units of 10,000). We observe that in all cases, RicciKGE leads to faster convergence and consistently higher MRR, demonstrating its effectiveness as a geometric regularizer that adapts local metric structures throughout the embedding process.

### F.8 Limitations and Future Work

While RICCIKGE introduces a principled framework for learning curvature-aware knowledge graph embeddings via discrete extended Ricci flow, several open challenges remain.

**Static Graph Assumption and Structural Rigidity.**  Our current design assumes a fixed knowledge graph topology throughout the training, which is common in the KGE literature. However, real-world KGs are typically *incomplete* (e.g., missing latent but meaningful links) and *noisy* (e.g., spurious triples from automated extraction). Fixing the graph limits the expressiveness of curvature: areas with overly negative curvature may reflect data sparsity rather than intrinsic hierarchy, and spurious links with anomalous positive curvature may distort local flow dynamics. Moreover, recent advances in graph rewiring (e.g., AFR (Fesser & Weber, 2024)) show that *dynamically adjusting the graph structure using geometric signals* can significantly mitigate over-squashing and over-smoothing. Extending RICCIKGE to support *curvature-guided edge rewiring*—where triples with high model confidence are added to low-curvature bottlenecks, and suspicious ones are pruned—could allow joint optimization of embeddings and graph structure. This would enable the Ricci flow to operate not only as a regularizer but also as a topological refiner.

**Cost of Full-Graph Curvature Recalculation.**  Although RICCIKGE captures geometric dynamics by iteratively evolving the curvature, computing the exact Ricci curvature (especially discrete variants based on Ollivier or optimal transport) across the entire graph remains computationally demanding. This overhead becomes a bottleneck in large-scale or streaming settings. In this work, we adopt full-graph updates to ensure theoretical consistency with the discrete Ricci flow and to avoid introducing additional sources of approximation that could affect convergence analysis. However, it is well-known that Ricci curvature is sensitive only to local perturbations. This suggests that a *local curvature refresh mechanism*—where only the edges incident to significantly updated entities are recomputed—may provide a practical acceleration while preserving flow alignment. Moreover, stochastic curvature sampling (e.g., refreshing a random subset of edges at each iteration) can further amortize the cost over training. Although these ideas are orthogonal to our method and conceptually simple to implement, they require careful design to avoid loss of geometric fidelity or optimization instability. We leave their systematic integration and analysis to future work.

**Scoring Function Compatibility.** Our current formulation adopts a unified scoring representation of the form $d(f_r(E(h)), E(t))$, which consists of a wide class of translational and geometric embedding models. Although this may seem restrictive, we emphasize that this form is not essential to the underlying optimization or evolution of the curvature of our framework. In fact, our coupling scheme between gradient dynamics and Ricci flow operates at the level of embedding updates and energy dissipation, and our theoretical convergence analysis does not depend on the specific algebraic structure of the scoring function. A long-term goal is to develop a unified curvature-guided embedding framework that encompasses a broad spectrum of KGE models while preserving geometric interpretability.

Together, these limitations highlight both the scope and the potential of geometry-aware KGE. Addressing them would enable RICCIKGE to serve not only as a curvature-aligned embedding method but also as a general-purpose tool for jointly evolving graph geometry, structure, and semantic representations.

## G  USAGE OF LLM

In preparing this work, we made limited use of ChatGPT (OpenAI) as an auxiliary tool. Its use was restricted to the following aspects:

- **Coding assistance**: ChatGPT-4o was occasionally consulted during debugging to suggest potential fixes for coding errors. All code was ultimately written, tested, and validated independently by the authors.
- **Language editing**: At the final stage of manuscript preparation, ChatGPT-5 was used to polish the English writing of the appendix. All suggestions were critically reviewed and adapted by the authors to preserve technical accuracy and consistency.

No AI tool was involved in formulating research ideas, designing or conducting experiments, or drawing scientific conclusions. All intellectual contributions are solely those of the authors.

