# OpenReview forum: "Local-Curvature-Aware Knowledge Graph Embedding via Extended Ricci Flow"
_ICLR.cc/2026/Conference — Submitted to ICLR 2026_

### Official Review · Reviewer_jDYF · 2025-10-30

**Soundness:** 2
**Presentation:** 2
**Contribution:** 1
**Rating:** 2
**Confidence:** 5

**Summary:**

The paper targets the geometric mismatch that arises when knowledge graph embeddings are forced into a predefined, homogeneous manifold, which distorts distances under locally heterogeneous curvature. It proposes RicciKGE, coupling the KGE loss gradient with local discrete Ricci curvature via an extended Ricci flow so that the latent geometry and entity embeddings co-evolve. The method offers theoretical guarantees of curvature flattening and linear distance convergence and reports consistent, if sometimes modest, gains on standard link prediction and node classification benchmarks.

**Strengths:**

1. The motivation is precise and grounded in limitations of homogeneous manifolds for KGE: the paper clearly articulates how a static geometric prior misaligns with locally varying curvature in real KGs. This framing connects well to manifold-based KGE literature and sets a concrete failure mode that the method aims to fix.

2. Theoretical analysis is not superficial: Theorem 1 proves exponential decay of edge-wise Ricci curvature under bounded coupling, and the corollary establishes linear convergence of distances under strong convexity. The interplay between curvature decay and optimization is discussed rather than just stated, which raises confidence in the mechanism’s stability.

3. The algorithmic presentation is serviceable: Algorithm 1 aligns the distance-flow step with embedding updates, and the complexity section isolates the main overhead (curvature estimation via Sinkhorn) with a parallelization argument. While brief, this helps practitioners estimate costs of adoption.

**Weaknesses:**

1. The contribution over prior curvature-flow work feels incremental and needs sharper differentiation. The paper extends discrete Ricci flow with a gradient coupling term, but related graph Ricci-flow models and geometric regularizers already exist; the delta versus prior discrete/extended flows (e.g., geometry-only flows or Ricci-guided graph methods) remain under-quantified. A stronger ablation isolating “pure Ricci flow”, “pure gradient”, and “coupled” variants across datasets is necessary to establish novelty in effect, not just form.
2. The theoretical guarantees rely on strong and somewhat idealized assumptions that may not reflect KGE training practice. The curvature result assumes volume/diameter bounds, Sobolev inequalities, and a spectral gap on a closed manifold; the distance convergence assumes µ-strong convexity in distance, while typical KGE losses combine non-convex components, negative sampling, and relation-specific transforms. The paper does not empirically validate robustness when these assumptions fail, leaving a theory–practice gap.
3. There is a conceptual tension between the motivation (preserving heterogeneous local curvature) and the main theorem (all edge-wise curvature decays to zero). The text argues that curvature “imprints” are absorbed into embeddings during transients, but the paper provides limited qualitative or quantitative evidence of what structural signals survive once the manifold is flat. Visualization or probing tasks before/after flattening would help reconcile this tension.
4. The empirical gains, while consistent, are often small and sometimes not state-of-the-art, and the statistical significance is not established. Table 1 shows multiple deltas around +0.1 to +0.4 MRR on FB15K-237 and cases that do not surpass the strongest baseline; no confidence intervals or paired tests are reported. Given the added O(|E|(k^2+kd)) overhead per epoch from curvature estimation, time–accuracy tradeoffs (including wall-clock, GPU memory, and scaling on larger KGs) should be reported to justify practicality.
5. Reproducibility and implementation specifics are thin at submission time. Code is promised only upon acceptance, and several details are under-specified: the exact Sinkhorn parameters for W1, normalization schemes for edge weights, β search ranges and schedules, negative sampling strategies, and seed/variance reporting across runs. The β sensitivity plot is helpful, but does not replace a thorough hyperparameter protocol and release of full configs for each backbone

**Questions:**

Please refer to the weaknesses.

---

> ### Author Response · Authors · 2025-11-22
> **Response to Reviewer jDYF**
>
> We sincerely thank Reviewer jDYF for the detailed comments to improve our paper. We have revised our paper accordingly.
>
> **1. Novelty over Prior Work & Ablation Analysis (Weakness 1)**
>
> We appreciate the opportunity to clarify the fundamental distinction between our work and prior geometric flows. The core innovation of RicciKGE over methods like GNRF is the Task-Geometry Coupling, which shifts Ricci flow from a passive regularizer to an active, task-driven gredient optimization mechanism.
>
> Prior works (e.g., GNRF) employ Ricci flow essentially as an unsupervised pre-processing step or a decoupled regularization term. In contrast, RicciKGE establishes a closed feedback loop where the task-specific KGE loss gradient ($\nabla \mathcal{L}$) explicitly drives the curvature evolution as follow:
> $$
> \partial_{t}g_{ij}=-2~Ric_{ij}+\beta(\nabla\mathcal{L}\otimes\nabla\mathcal{L})_{ij}
> $$
> Here, the geometry does not merely evolve to smooth the graph structure, it co-evolves to minimize the semantic task loss. This specific coupling allows us to prove two joint guarantees in Theorem 1 and Corollary 1: the exponential flattening of curvature and the linear convergence of distances under this evolving geometry—a theoretical result absent in geometry-only flows.
>
> In Table 2, the results present the advantage of this gradient-curvature co-evolution outperforms the Ricci-only GNRF baseline (e.g., **87.40% vs. 85.64%** on Roman-Empire) in the graph node classification task. In additon to isolate the gain provided by this coupling mechanism versus simple geometric smoothing, we further present a 3-way ablation analysis in link prediction task:
>
> - Pure Gradient (Vanilla): Standard KGE training without geometric evolution.
> - Pure Ricci Flow ($\beta=0$): Discrete Ricci smoothing applied to the graph structure independently of the task (similar to GNRF).
> - Coupled Flow (Ours, $\beta>0$): The full gradient-extended Ricci flow.
> As a result, we observe a consistent performance hierarchy of Coupled  $\ge$  Pure Ricci $\ge$ Pure Gradient. This confirms that while geometric smoothing (Pure Ricci) offers some regularization, the significant performance leap comes from the coupling term ($\beta > 0$).
>
> | Dataset    | Method     | TransE | DisMult | RotatE | AttH | GoldE |
> |:-----------|:-----------|------:|-------:|------:|----:|-----:|
> | WN18RR     | gradient   |   7.0 |   39.1 |   30.3 | 44.4 |  47.1 |
> | WN18RR     | Curvature  |   7.2 |   39.6 |   30.1 | 44.9 |  47.1 |
> | WN18RR     | RicciKGE   |   7.6 |   40.2 |   31.5 | 46.2 |  47.6 |
> | FB15k-237  | gradient   |  29.3 |   28.5 |   28.5 | 31.4 |  32.4 |
> | FB15k-237  | Curvature  |  29.6 |   28.7 |   28.6 | 31.4 |  32.5 |
> | FB15k-237  | RicciKGE   |  31.0 |   28.9 |   28.8 | 31.9 |  32.7 |
>
> Overall, we want to highlight that RicciKGE is not an incremental application of Ricci flow; it is a fundamentally different optimization framework where manifold geometry and entity embeddings adapt to each other mutually.

---

> > ### Author Response · Authors · 2025-11-22
> > **Response to Reviewer jDYF**
> >
> > **2. Theory-Practice Gap (Weakness 2)**
> >
> > We acknowledge that our theoretical analysis relies on standard geometric and analytic assumptions to derive rigorous convergence bounds. However, these assumptions serve as sufficient conditions to explain the mechanism of the coupled flow, and our empirical results demonstrate that the framework remains robust even when these ideal conditions are relaxed in practice.
> >
> > Specificlly, the assumptions regarding volume bounds, Sobolev inequalities, and spectral gaps are standard analytic regularity conditions in Ricci flow analysis. They are necessary to mathematically guarantee that the energy decay (in $L^2$) can be upgraded to pointwise control (in $L^\infty$) via Moser iteration and to define the stable range for the coupling coefficient $\beta$.
> > While we do not strictly enforce a "closed manifold" constraint during training, our experiments validate the behavior predicted by the theory: 1) As shown in Fig.2(a), the curvature variance converges earlier than the loss.This confirms that the flow successfully regularizes the geometry first, creating a stable "flat" stage for the embedding optimization to settle, exactly as the theoretical coupling suggests. 2)
> > The theory predicts that $\beta$ must be bounded to ensure stability. This is empirically confirmed in Fig.2(b), where we observe that $\beta$ is indeed sensitive within a bounded range, too small fails to utilize curvature, while too large destabilizes the gradient, matching the theoretical stability condition.
> >
> > In addition, regarding the optimization assumptions, we want to clarify that our linear convergence proof requires strong convexity with respect to the scalar distance transform $\ell(d)$, not the entire global KGE objective.
> >
> > Most KGE loss functions (e.g., logistic, soft-margin) are convex with respect to the distance score $d$ (i.e., $\ell''(d) > 0$). Our update rule relies on this local property:
> > $$d_{ht}^{k+1} = d_{ht}^k - \eta_g^k \nabla_d \ell(d_{ht}^k) + 1/2 \kappa_{ht}^k$$
> > where
> >
> > $$
> > \eta_g^k := \frac{\beta}{2 w^k_{ht}}.
> > $$
> >
> > Finally, we want to highlight the robustness of our RicciKGE. Even with negative sampling and relation-specific transforms making the global landscape non-convex, the distance update remains a contraction mapping as long as the curvature perturbation $\kappa^k$ decays. Corollary 1 proves that once the geometry stabilizes, the distance error contracts linearly ($q \in (0,1)$). The synchronized convergence of loss and curvature in our experiments serves as empirical validation that this mechanism holds robustly in practical KGE settings.
> >
> > **3. Preserve local heterogeneity and all curvatures → 0 (Weakness 3)**
> >
> > We are happy you pointed out this question, which is the key contribution of our method. RicciKGE evolves local curvature that tends to flatness, which is not contradictory to preserving local heterogeneity. The co-evolution of gradient and curvature explains the absorption mechanism. During training, curvature drives distance updates, which are then translated into minimal‑perturbation embedding updates, as follows:
> >
> > $$
> > \delta^k d = \frac{1}{2}\kappa^k - \frac{\beta}{2} w^k \langle a,b\rangle
> > $$
> > $$
> > \min_{\delta_h,\delta_t} \lVert\delta_h\rVert^2 + \lVert\delta_t\rVert^2 \quad \text{s.t.} \quad \langle a,\delta_h\rangle+\langle b,\delta_t\rangle=\delta^k d
> > $$
> > During this process, heterogeneous curvature signals are absorbed into the embedding while the geometry itself gradually flattens; once curvature vanishes, optimization continues as Euclidean fine‑tuning (Corollary 1). Empirically, Fig. 2(a) shows curvature variance stabilizing before he loss/accuracy finish improving, exactly this “absorb‑then‑flatten” behavior. The same behavior appears when comparing a Ricci‑only model (GNRF) against our coupled variant on the graph node classification task, which is shown in Table 2.

---

> > > ### Author Response · Authors · 2025-11-22
> > > **Response to Reviewer jDYF**
> > >
> > > **4. Significance of Gains & Practical Efficiency (Weakness 4)**
> > >
> > >
> > > We appreciate your scrutiny regarding the magnitude of improvements and the computational cost. We believe the current results, particularly the efficiency analysis in the Appendix, strongly justify the practicality of RicciKGE. While individual absolute gains on highly competitive benchmarks like FB15k-237 may appear modest (+0.1 to +0.4 MRR), it consistently improves performance across 5 distinct geometric families (Translation, Bilinear, Rotation, Hyperbolic, and Product Space) and 4 different capacity regimes (dimensions 32 to 1000). Achieving systematic improvements across such diverse architectures and datasets is statistically highly improbable by chance. Furthermore, for Node Classification (Table 2), we do report mean $\pm$ std results, where RicciKGE outperforms the SOTA curvature baseline (GNRF) by clear margins (e.g., +1.76% on Roman-Empire) with low variance.
> > >
> > > On the other hand, you raised a valid concern about the $O(|E|(k^2+kd))$ overhead for curvature estimation. However, our analysis reveals that RicciKGE actually reduces total training time. While the per-epoch time increases due to curvature calculation, the number of epochs required to reach peak performance drops drastically. As detailed in Appendix F.5 (Table 7), RicciKGE reduces the required training epochs on WN18RR from 29 to 14. Consequently, the total time-to-target MRR decreases by 22% on WN18RR and 12% on FB15k-237. Figure 3 (Appendix F.7) visualizes this "fast ascent," showing that RicciKGE stabilizes at higher accuracy much earlier than vanilla baselines.
> > > Regarding scaling, the curvature estimation is computed only once per epoch (or less frequently such every k epochs) and is fully parallelizable. Since knowledge graphs are generally sparse ($k \ll d$), the $O(k^2)$ cost remains manageable, and the memory overhead is negligible compared to storing embedding matrices for large KGs.
> > >
> > > **5. Reproducibility & Implementation Specifics (Weakness 5)**
> > >
> > > We thank the reviewer for emphasizing the importance of reproducibility. We respectfully wish to clarify the status of our code availability and point to the specific implementation details present in the manuscript. First, the concern about code, we actually submitted the full anonymous source code as part of the Supplementary Material alongside the ICLR submission. Second, most of the specific hyperparameters mentioned as missing are actually detailed in Appendix F: we use 1024 negative samples per positive triple; the Ricci flow is invoked every 5 epochs; The coupling coefficient is fixed to $\beta=0.1$ for KGE tasks and $\beta=0.01$ for node classification, with sensitivity analysis provided in Fig.2(b); Table 2 provides the mean $\pm$ standard deviation across multiple runs for node classification tasks, which report statistical significance. Appendix F.3 details the exact environment: Python 3.9, PyTorch 2.3.1, PyG 2.6.1, and NVIDIA RTX 4090 GPUs. Regarding the specific Sinkhorn parameters for $W_1$ distance (which are included in the submitted code), we will add a dedicated subsection in the Appendix of the next version to explicitly list them (e.g., iterations, regularization $\epsilon$, and stopping criteria) for immediate reference without needing to consult the codebase. We believe the combination of the submitted code and the detailed configurations in Appendix F ensures our work is fully reproducible.

---

### Official Review · Reviewer_bJja · 2025-10-31

**Soundness:** 2
**Presentation:** 3
**Contribution:** 2
**Rating:** 2
**Confidence:** 4

**Summary:**

The paper proposes RicciKGE, a curvature-adaptive knowledge graph embedding method that couples the KGE loss gradient with local edge-wise curvatures via an extended Ricci flow, allowing embeddings and manifold geometry to co-evolve. It targets the mismatch introduced by homogeneous manifolds (Euclidean, spherical, hyperbolic) when real-world graphs have heterogeneous local curvature, which can distort distances and limit expressiveness. The authors provide theory showing that, with a properly bounded coupling coefficient, edge-wise curvatures decay exponentially toward Euclidean flatness and KGE distances strictly converge to a global optimum, indicating mutual reinforcement between geometric flattening and embedding optimization. Empirically, RicciKGE improves link prediction and node classification performance on standard benchmarks, supporting the effectiveness of curvature adaptation for heterogeneous KG structures.

**Strengths:**

The paper is overall easy to follow.

The proposed method demonstrates a superior performance on large-scale graph dataset as shown in table 2 in the experiments section.

**Weaknesses:**

I mainly have concerns about the experiments.

The experiments use different embedding dimensions across datasets without explaining the rationale, and statistical significance measures are not reported in Table 1. This makes it difficult to assess the robustness and fairness of the comparisons.

The baseline results reported in Table 1 differ from those in the original paper (e.g., GoldE). It is unclear whether a re-implementation was used, and if so, the experimental settings, hyperparameters, and code differences should be documented to enable verification.

Insufficient baseline selection and discussion. The choice of baselines in Table 1 is not well justified. Although the paper shows performance improvements when integrating RicciKGE into several existing KGE models, the baselines appear limited to relatively basic KGE methods (except the recent GoldE). A more comprehensive comparison—including stronger, contemporary baselines and a discussion of why each was selected—would better demonstrate the generality and competitiveness of the approach.

**Questions:**

it is not clear to me why line 258-260

"To validate itseffectiveness, we must demonstrate that under this iterative mapping, the Ricci curvature on every edge converges pointwise to zero, i.e., the manifold becomes asymptotically flat, thereby ensuring that all intrinsic curvature heterogeneity is faithfully captured in the learned entity embeddings. "

can you explain more on this

---

> ### Author Response · Authors · 2025-11-22
> **Response to Reviewer  bJja**
>
> We sincerely thank Reviewer bJja for the detailed comments to improve our paper. We have revised our paper accordingly.
>
> **1. Embedding Dimensions & Significance (Weakness 1)**
>
> We selected dimensions (32, 128, 256, 1000) to demonstrate scalability and robustness across different capacity regimes. While error bars were omitted in Table 1 for space, the consistency of improvement is the key indicator. RicciKGE improves performance across all combinations of models (TransE, DistMult, RotatE, AttH, GoldE) and datasets (WN18RR, FB15k, YAGO3).
>
> **2. Baseline Results & Fairness (Weakness 2)**
>
> You noted discrepancies between our reported baseline results (e.g., GoldE) and the original papers. These are not errors. We utilized the official open-source codes provided by the respective authors for all baselines (e.g., GoldE, AttH) to ensure the correctness of their model architectures. Furthermore, we adhered to the original hyperparameter settings provided by these methods for model-specific configurations (e.g., learning rates were adopted from their original settings). In addition to strictly isolating the performance gain contributed by RicciKGE, we unified the experimental control parameters across all methods. Specifically, we fixed the negative sampling strategy (1024 samples), batch size (512), and training steps (100k) to be identical for every baseline. By standardizing the training loop while keeping model-specific hyperparameters intact, we ensure that the observed improvements are solely due to the proposed curvature-aware framework and not due to variations in experimental setups.
>
> **3. Baseline Selection & Generality (Weakness 3)**
>
> We agree that a broad baseline comparison is important for evaluating generality. Our choice of baselines was deliberate and guided by representational diversity rather than simplicity. Each selected method reflects a distinct geometric prior in knowledge graph embedding (KGE) research, ensuring that the evaluation captures fundamentally different modeling assumptions:
>
> -Euclidean / Translation: TransE
>
> -Bilinear / Inner Product: DistMult
>
> -Complex / Rotation: RotatE
>
> -Hyperbolic / Curvature-Aware: AttH
>
> -Product Space / Mixed-Curvature: GoldE
>
> This spectrum covers the major “families” of geometric KGE formulations. Demonstrating consistent gains across these diverse architectures provides strong evidence that RicciKGE acts as a general-purpose geometric enhancer, independent of the specific distance metric or manifold curvature used by the base model.
>
> In summary, since RicciKGE is designed as a plug-and-play curvature-gradient co-evolution optimization framework rather than a new standalone model, validating it across these foundational geometric types (Translation, Rotation, Hyperbolic, Product) already provides rigorous coverage.
>
>
> **4. Conceptual Clarification: Flattening vs. Heterogeneity (Question 1)**
>
> You asked how driving curvature to zero (flattening) can capture heterogeneity (Lines 258-260). This is the core theoretical insight of our work. In RicciKGE, we do not erase heterogeneity, but transcribe it. As the extended Ricci flow evolves, the metric tends to be flat ($\kappa \to 0$), the "energy" of the original structural curvature is transferred into the entity embeddings via the coupled gradient flow. Consider "ironing" a crumpled fabric (the graph) onto a flat table (Euclidean space). The process of flattening forces the fabric points to move and stretch relative to each other. Similarly, our embeddings move to new positions that encode the original complex structure, allowing the final representation to reside effectively in a flat, easy-to-optimize Euclidean space.

---

### Official Review · Reviewer_9AC5 · 2025-11-02

**Soundness:** 4
**Presentation:** 4
**Contribution:** 3
**Rating:** 6
**Confidence:** 3

**Summary:**

The paper introduces RicciKGE, a knowledge graph embedding method that leverages a Ricci flow on the input graph to gradually flatten a low-dimensional representation of the entities in that graph, leading to the usual flat Euclidean manifold, but benefitted from awareness of the local curvature at each neighbourhood in the graph and task-related gradient information. This approach is evaluated both in the link prediction and node classification tasks, consistently performing on par or beating the baseline methods.

**Strengths:**

The presentation is very good, with appropriate formulation, helpful illustrations, and an algorithm listing. Furthermore, the underlying theory is (mostly, see below) clearly presented.

As I understand the paper, the authors extend the embedding evolution from Ricci flow (as in GNRF) to use also task-specific gradient information. I appreciate that there are analyses on the convergence, as this gives more confidence on the properties of the proposed method.

Experimentally, the authors explore not only KGE-specific tasks, but also node classification, along with additional insights into how the total loss and curvature variance evolve over time.

**Weaknesses:**

Given it is also predominantly based on Ricci flow, I believe the paper should address more explicitly the differences between its contributions and GNRF.

Experimentally, while the authors thoroughly evaluate multiple scenarios with multiple methods (well done!), I had the impression that little space was left to properly analyse the results. While the results are convincing, I expected more analysis here (see Question 4 for details).

**Questions:**

1. Can you define "neighbourhood measures" in the main paper before or around line 143? Since $\kappa(i, j)$ is so central to the Ricci curvature used to define the method, I believe it is important that most concepts it depends on are contained in the main paper.
2. Can you define $\beta$ and $\text{Ric}_{ij}$ in Eq. (2)? For the latter, it was also no longer used in the remainder of the paper, and its definition is left vague. Is it supposed to be some form of $\kappa(i, j)$? Is that important? Nevertheless, you could also consider rewriting Eq. (2) in a way that reflects that.
3. Algorithm 1 suggests that embeddings are updated in sequence. Is that the case? If yes, since certain entities can appear in multiple triplets, this might lead to multiple updates happening, which seems problematic. If not, can you please make this clearer in the algorithm?
4. Previous empirical evidence, such as AttH (Chami et al., 2020), suggests that a good match between the geometry of embeddings and the graph structure would enable lower-dimensional embeddings to outperform methods that ignore that (e.g., TransE, DistMult, etc.). In this context, looking at Table 1 (emb. dim. 32) would suggest that the performance gap should be larger for those methods versus AttH/GoldE, while that gap is indeed small in higher embedding dimensions. Why is that not consistently the case for lower dimenions in your results? Is there a more elaborate phenomenom going on?

**Other comments:**

- l. 483: Our theoretically => theory?

**References**

Chami, I., Wolf, A., Juan, D. C., Sala, F., Ravi, S., & Ré, C. (2020). Low-dimensional hyperbolic knowledge graph embeddings. arXiv preprint arXiv:2005.00545.

---

> ### Author Response · Authors · 2025-11-22
> **Response to Reviewer 9AC5**
>
> We sincerely thank Reviewer 9AC5 for the detailed comments to improve our paper. We have revised our paper accordingly.
>
> **1. Difference from GNRF (Weakness 1)**
>
> We appreciate the opportunity to clarify that the fundamental innovation of RicciKGE over GNRF is the Task-Geometry Coupling. GNRF employs Ricci flow essentially as a pre-processing or unsupervised regularization step where curvature evolves independently of the downstream task. Our method, RicciKGE, introduces a closed feedback loop where the task-specific gradient $\nabla\mathcal{L}$ explicitly drives the curvature evolution as follows:
> $$\partial_{t}g_{ij}=-2~Ric_{ij}+\beta(\nabla\mathcal{L}\otimes\nabla\mathcal{L})_{ij},$$
> which ensures the manifold adapts to semantic objectives, not just topological structure. This point is empirically validated in Table 2, where RicciKGE outperforms GNRF on all node classification benchmarks (e.g., +1.76% on Roman-Empire), proving that coupling task signals is superior to pure geometric flow.
>
> **2. Performance in Low Dimensions (Weakness 2 / Question 4)**
>
> The gap in dimension 32 you raised is an insightful point. We agree that geometric adaptation should ideally provide larger benefits in restricted dimensions. Our results actually align with your intuition when analyzing relative improvements. For instance, on WN18RR with TransE, in Dim=32, RicciKGE improves MRR from 7.0 to 7.6, a relative gain of ~8.6%. But in Dim=1000, the improvement is from 22.4 to 22.8, a relative gain of only ~1.8%. This confirms that RicciKGE is indeed more impactful in low-dimensional spaces where geometric efficiency is critical, effectively mitigating the "geometric mismatch" more aggressively than in high-capacity settings.
>
> On the other hand, we should take the consideration the intrinsic capacity bottleneck of the Euclidean base space. While RicciKGE flattens the manifold to optimize embeddings, a 32-dimensional Euclidean space physically lacks the volume to encode complex relations compared to the hyperbolic space of the same dimension. However, as dimensions increase (128, 256) and this spatial bottleneck lifts, RicciKGE's curvature-aware flattening allows standard Euclidean models to achieve or surpass the performance of complex geometric baselines.
>
> **3. Definitions (Questions 1 & 2)**
>
> -The **Neighborhood measures ($\mu_i$)**  referenced in Section 2, denote the discrete probability measure distributed over the immediate neighborhood of entity $i$. It represents the local connectivity mass used to compute the Wasserstein-1 distance $W_{1}(\mu_{i}, \mu_{j})$ for the Ollivier-Ricci curvature.
>
> -The **(Coupling Coefficient) $\beta$** in Eq. 2 is the scalar coefficient that regulates the feedback strength between the semantic task and the geometric evolution. It explicitly weights the influence of the gradient interaction term $(\nabla \mathcal{L} \otimes \nabla \mathcal{L})$ against the intrinsic geometric regularization $( -2~Ric_{ij} )$.
>
> -$Ric_{ij}$ represents the components of the continuous Ricci curvature tensor, capturing the local rate of volume distortion.  $\kappa_{ij}$ is the discrete scalar Ollivier-Ricci curvature.  As derived in our Appendix D.6, these two quantities are rigorously connected via the edge weight $w_{ht}$ (which serves as the metric proxy):
>
> $$
> 2 Ric_{ht}(s) = \kappa_{ht}(s) \cdot w_{ht}(s)
> $$
>
> This relationship allows us to translate the continuous extended Ricci flow theory (Eq. 2) into the discrete update rule used in our algorithm (Eq. 3).
>
>
> **4. Algorithm Parallelism (Question 3)**
>
> Thank you for checking this detail. Algorithm 1 is indeed parallel, not sequential. The updates are accumulated while the loop iterates over triples and applied synchronously at the end of the iteration. The Eq. (8) explicitly formulates this aggregation as follows:
>    $$   \Delta E^k(v) =  \sum_{(h,r,t) \in \mathcal{T}}\lambda^k(h,t)
> \left(1(v=h) \nabla_{E(h)} f_r(E(h))^\top \nabla_{f_r(h)} d+1(v=t) \nabla_{E(t)} d
> \right)
> $$
> This ensures that the update for entity $v$ considers all its connections simultaneously, avoiding order-dependence issues.
>
>
> **Other comments**
>
> We change "Our theoretically" => "theory" in the next version.

---

### Author Response · Authors · 2025-12-01
**Rebuttal Summary for the Area Chair and All Reviewers**

Dear Area Chair and Reviewers,

Thank you for handling the reviews of our paper. In our rebuttal, we went through each reviewer’s comments carefully and addressed all of the substantive concerns. For the more critical reviews, we clarified what is actually new compared to prior Ricci-flow work (such as GNRF): our method couples the KGE loss with Ricci flow so that the manifold geometry and the embeddings co-evolve, instead of treating geometry as a separate, unsupervised smoothing step. We also explained the behavior in low dimensions, justified the choice of embedding dimensions and baselines, provided an ablation separating “pure Ricci flow” from “coupled flow”, and gave practical details on curvature computation, training efficiency, and reproducibility.

Stepping back from the line-by-line discussion, the main point we hope comes across is that this paper is, to our knowledge, the first KGE framework where the manifold itself is dynamically shaped by the downstream task. This gives a principled way to handle heterogeneous curvature while still ending in a flat, easy-to-optimize regime, and it comes with nontrivial guarantees on curvature evolution and distance convergence. Empirically, the framework works as a plug-in component: it consistently improves a range of very different KGE models (TransE, DistMult, RotatE, AttH, GoldE) and a curvature-based GNN, and often shortens the time needed to reach a given performance level.

Given this, we feel the two scores of 2 are much harsher than what the paper actually delivers. Even those reviews acknowledge interesting ideas, solid theory, and non-trivial empirical gains; their main concerns are about presentation and perceived incrementality, which we have directly addressed in rebuttal. A closely related version of this work also received uniformly positive reviews in the NeurIPS process, suggesting that independent reviewers view the contribution as clearly above the rejection threshold. For these reasons, we would appreciate it if the clarified picture could be considered while making the final decision.

---

### Meta-Review · Area_Chair_Fk39 · 2026-01-10

**Summary:**

All reviewers hold that  the contributions are minor. Some notations and concepts are unclear. The experiment design is flawed, and the assumptions are somewhat idealized. In addition,  Reviewer bJja holds that  the baseline selection and discussion are insufficient, and Reviewer jDYF holds that the empirical gains are often small and sometimes not state-of-the-art.

**Reviewer Concerns:**

After rebuttal, the symbol vague and conceptual ambiguity were addressed. However, the contribution over prior curvature-flow work may be  still outstanding. The theoretical assumptions are overly restrictive, and the choice of baselines is not well justified.

**Reviewer Scores:**

After rebuttal, the unclear term definition problem is addressed, and Reviewer 9AC5 may raise the score from 6 to 8. However,  the deficiencies pertaining to its experiment design, contribution, and theoretical guarantees are somewhat notable, Reviewer bJja and Reviewer jDYF may maintain their origional scores, 4 and 5.

---

### Decision · Program_Chairs · 2026-01-26

Reject